# ACTIVE TASK DISAMBIGUATION WITH LLMS

**Katarzyna Kobalczyk**,\* **Nicolás Astorga**\*, **Tennison Liu, & Mihaela van der Schaar**
DAMTP, University of Cambridge
{knk25, nja46}@cam.ac.uk

## ABSTRACT

Despite the impressive performance of large language models (LLMs) across various benchmarks, their ability to address ambiguously specified problems–frequent in real-world interactions–remains underexplored. To address this gap, we introduce a formal definition of task ambiguity and frame the problem of task disambiguation through the lens of Bayesian Experimental Design. By posing clarifying questions, LLM agents can acquire additional task specifications, progressively narrowing the space of viable solutions and reducing the risk of generating unsatisfactory outputs. Yet, generating effective clarifying questions requires LLM agents to engage in a form of meta-cognitive reasoning, an ability LLMs may presently lack. Our proposed approach of active task disambiguation enables LLM agents to generate targeted questions maximizing the information gain. Effectively, this approach shifts the load from implicit to explicit reasoning about the space of viable solutions. Empirical results demonstrate that this form of question selection leads to more effective task disambiguation in comparison to approaches relying on reasoning solely within the space of questions.

## 1 INTRODUCTION

Recent advances in the field of LLMs have led to the development of problem-solving agents capable of addressing complex tasks that extend far beyond conventional structured data problems such as regression and classification. State-of-the-art LLMs have demonstrated remarkable success in logical reasoning (Creswell et al., 2022; Lei et al., 2023), mathematical problem solving (Romera-Paredes et al., 2024; Imani et al., 2023), code generation (Liu et al., 2024; Zhang et al., 2023a) or creative writing (Coenen et al., 2021; Chakrabarty et al., 2023). While existing research predominantly focuses on enhancing LLMs' planning and reasoning capabilities with new prompting strategies like

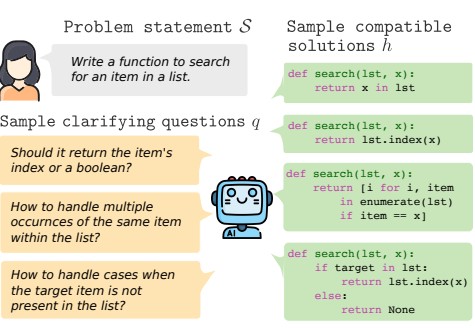

Figure 1: An ambiguous problem statement, a sample of LLM-generated compatible solutions, and clarifying questions. ▶**The goal:** Generate the most informative question.

Chain of Thought (CoT) (Wei et al., 2022) or self-consistency (Wang et al., 2023), evaluation benchmarks typically assume complete and unambiguous problem statements. However, due to the inherent ambiguity of natural language (Stengel-Eskin et al., 2023; Liu et al., 2023) or deliberate underspecification, tasks encountered in real-world usage of LLMs may often not be well-defined, increasing the risk of the agent misinterpreting the true intentions of the problem setter.

The concurrent line of work suggests that in the presence of ambiguously specified tasks, agents should be able to infer missing information to discern the intended behavior (Tamkin et al., 2023). However, in the context of human-specified tasks, when user intentions deviate from that of the average population on which the internal LLM preference model has been trained, the agent is at risk of generating outputs that do not align with the user's true needs. Such behaviors may be especially harmful in safety-critical applications, such as medical diagnosis or treatment decisions, where erroneous answers pose significant risks.

---

\*Equal contribution

Similar to prior works (Kuhn et al., 2022; Li et al., 2025), we advocate for LLM agents to engage in an interactive dialogue when faced with ambiguously defined problems. By asking clarifying questions, agents can narrow down viable solutions, reducing the risk of generating unsatisfactory responses. Li et al. (2025) have demonstrated that such interactive task elicitation can be more informative and require less effort than refining the original prompts, often uncovering novel, not initially anticipated aspects of the given problem. However, asking a good clarifying question requires LLM agents to reason about their own generative distribution and identify features that differentiate the viable solutions. We hypothesize that such meta-cognitive skills (Kuhn et al., 2022; Barzilai & Zohar, 2016) are beyond the capabilities of many modern LLMs. In response, this paper investigates whether out-of-the-box abilities of LLMs in asking clarifying questions can be improved.

While traditional machine learning has extensively explored efficient data collection under model uncertainty, resulting in the development of active learning strategies (Houlsby et al., 2011; Mackay, 1992), LLM-based agents do not operate on well-defined and structured input-output spaces. Instead, LLMs navigate complex tasks with natural language, necessitating novel approaches to efficient information acquisition. Drawing from the principles of Bayesian Experimental Design (BED), we propose a method of active task disambiguation maximizing questions' utility by direct estimation of their information gain. Effectively, the proposed method shifts the load from implicit reasoning about the best question to explicit reasoning via sampling from the solution space.

**Contributions:** ▶ We identify the need for new reasoning methods enabling LLM agents to handle ambiguously specified tasks effectively. ▶ Section 2: We introduce a formal definition of task ambiguity in natural language problem specifications, enabling us to frame the problem of effective task disambiguation through the lens of BED. ▶ Section 3: We propose and motivate theoretically a BED-based strategy for LLM agents to generate clarifying questions. ▶ Section 4: We evaluate the effectiveness of competing question-generating strategies on an illustrative game of 20 questions and a real-world application to code generation. Our results demonstrate that the BED-based method improves upon baseline strategies relying on implicit reasoning about the best question to ask.

## 2 FORMALISM & BACKGROUND

We let $\Sigma$ denote the space of natural language. We define a problem statement $\mathcal{S} \in \Sigma$ as a natural language instruction for an agent to generate a solution $h \in \Sigma$ belonging to the unknown set of ground-truth solutions $\mathcal{H}^* \subset \Sigma$. We assume that the problem statement, $\mathcal{S}$, can be decomposed into two parts: a set of requirements $\mathcal{R}$ that any $h \in \mathcal{H}^*$ should satisfy, and any additional contextual information $\mathcal{C}$ that may influence the preference towards different outputs, $h \in \Sigma$.

**Example 1** (Code generation). Consider the problem of code generation with an LLM agent based on a prompt $\mathcal{S} = (\mathcal{R}, \mathcal{C})$. Here, $\mathcal{C}$ is the natural language instruction guiding the LLM to output a code solution $h$. The requirements for the generated code, described in $\mathcal{R}$, include the expected functionality of the code and any additional test cases that the generated solutions should pass. The ground truth $\mathcal{H}^*$ contains all programs $h$ that, e.g. pass all hidden test cases or satisfy the internal needs of the human user. A concrete example is presented in Figure 1.

Let $p^*(\cdot|\mathcal{S})$ denote the likelihood function that determines the preferences of the problem setter over candidate solutions $h \in \Sigma$. We assume that for any $\mathcal{S}$, $p^*(\cdot|\mathcal{S})$ may be decomposed as:

$$p^*(h|\mathcal{S}) = \mathbb{1}\{h \vdash \mathcal{R}\}\tilde{p}^*(h|\mathcal{R}, \mathcal{C}), \qquad \forall h \in \Sigma. \tag{1}$$

In the above, $\mathbb{1}\{h \vdash \mathcal{R}\}$ is an indicator function assessing if a solution $h$ satisfies the requirements $\mathcal{R}$ or not. This indicator function is objective in the sense that it represents an unquestionable truth about a sample solution $h$. In contrast, $\tilde{p}^*(\cdot|\mathcal{R}, \mathcal{C})$, is context dependent and subjective. Given the same contextual information, different problem setters (e.g., different human users) may have varying preferences over competing solutions that satisfy the given set of requirements.

In this paper, we will focus on the problem of task underspecification; a situation where the support of $p^*(\cdot|\mathcal{S})$ extends beyond the set of acceptable solutions.

**Definition 1** (Task ambiguity). *Let $\mathcal{S} = (\mathcal{R}, \mathcal{C})$ and $\mathcal{H} := \{h : h \vdash \mathcal{R}\}$. We say that $\mathcal{S}$ is ambiguous if $\mathcal{H}$ is a proper superset of $\mathcal{H}^*$, i.e. $\mathcal{H} \supset \mathcal{H}^*$.*

When an LLM agent attempts to solve a given task specified by $\mathcal{S}$, it generates a solution $h$ according to its own generative distribution $p_{\phi_h}(\cdot|\mathcal{S})$. If $\mathcal{S}$ is ambiguous, providing a correct solution becomes

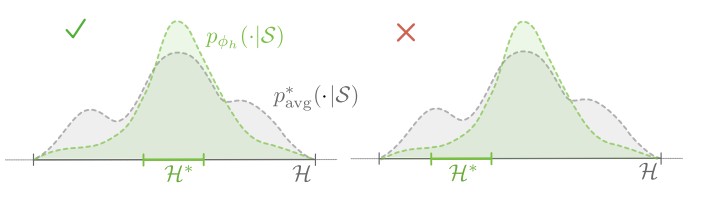
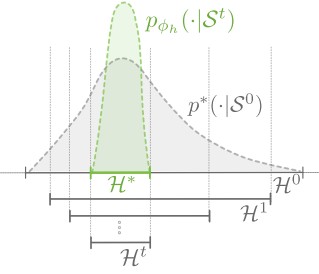

(a) *Preference optimization*. The distribution $p_{\phi_h}$ is aligned with the average preferences of the population $p^*_{\text{avg}}(\cdot|\mathcal{S})$. This strategy is successful, if the set of acceptable solutions $\mathcal{H}^*$ is aligned with the mode of $p_{\phi_h}$ (left). If not, $p_{\phi_h}$ may fail to generate correct answers (right).

(b) *Iterative task elicitation*. By acquiring new task specifications, the set of admissible solutions is successively reduced, surrounding $\mathcal{H}^*$.

Figure 2: *Resolving ambiguity*.

challenging due to the risk of misinterpreting the true intention of the problem setter. The risk of misinterpretation is high if the distributions $p^*$ and $p_{\phi_h}$ are not aligned. In order to solve this problem, one could consider at least two alternative approaches:

**Preference optimization.** The first school of thought postulates that agents should be able to fill in the blanks by combining information from instructions and their previous experience in order to identify the intended behavior (Tamkin et al., 2023). In particular, the goal of reinforcement learning with human feedback (RLHF) (Bai et al., 2022; Ouyang et al., 2022) or direct preference optimization (DPO) (Rafailov et al., 2023) is to align the generative distribution $p_{\phi_h}(\cdot|\mathcal{S})$ with preferences of the overall population such that with a high likelihood, $p_{\phi_h}(\cdot|\mathcal{S})$ concentrates around $\mathcal{H}^*$ (Fig. 2a, left). However, if the intentions of an individual user deviate from that of the average population (Fig. 2a, right), the agent is at risk of proposing solutions not belonging to $\mathcal{H}^*$. Such behaviors may be especially harmful if there is an associated risk with generating a sample answer $h \notin \mathcal{H}^*$.

**Iterative task elicitation.** The second approach involves engaging in an interactive dialogue with the user, seeking additional specifications (Kuhn et al., 2022; Li et al., 2025). Since $\mathcal{H} \supset \mathcal{H}^*$, some requirements of the problem must have not been explicitly stated in $\mathcal{S}$ and remain unknown to the LLM agent. Let $\mathcal{S}^t = (\mathcal{R}^t, \mathcal{C}^t)$ denote the task specification at time $t$. During each round of the conversation, the agent can ask a clarifying question, $q^t$, and after receiving the oracle answer $a^t$, the problem statement gets updated to $\mathcal{S}^{t+1} = \mathcal{S}^t \cup (q^t, a^t)$. At each interaction step, the set of solutions compatible with $\mathcal{S}^t$, $\mathcal{H}^t := \{h : h \vdash \mathcal{R}^t\}$, becomes smaller and successively concentrates around the true solution set $\mathcal{H}^*$, mitigating the risk of failure when sampling $h \sim p_{\phi_h}(\cdot|\mathcal{S}^t)$. The goal of this approach is to ensure personalized alignment without the need for model fine-tuning.

**The goal.** This paper focuses on the latter option of interactive task elicitation. Given that obtaining the oracle answers maybe be costly[1] , our goal is to find the best strategy for generating clarifying questions so that $p_{\phi_h}$ concentrates around $\mathcal{H}^*$ with minimal interaction cost.

> 💡 We make a distinction between model uncertainty and the ambiguity of a problem. While model uncertainty is measured through the heterogeneity of $p_{\phi_h}$, ambiguity of a task is defined through the objective indicator function $\mathbb{1}\{h \vdash \mathcal{R}\}$ whose true value is independent of the LLM or the problem setter. We note that in many cases, the support of the agent's generative distribution $p_{\phi_h}(\cdot|\mathcal{S})$ may cover a larger space than $\mathcal{H}^*$, even if the problem statement $\mathcal{S}$ is not ambiguous. This is due to the persistent issue of LLM hallucinations (Zhang et al., 2023b; Rawte et al., 2023) or the complexity of the problem exceeding the LLM's abilities. Thus, a high uncertainty of the LLM agent does not imply inherent ambiguity of $\mathcal{S}$. While the focus of this paper lies in addressing objectively ambiguous problems, as we will see in experiment 4.2, iterative task elicitation may also improve LLM-generated solutions even for unambiguous problems.

## 2.1 WHAT MAKES A QUESTION INFORMATIVE?

To understand what makes a question informative, we will rely on the principles of Bayesian Experimental Design (BED). BED aims to design optimal experiments by maximizing the amount of information about an unknown quantity of interest gained from an outcome of an experiment. In the context of problem solving with LLM agents, an experiment corresponds to a question $q$ posed

---

[1]We work under the convention of BED (Rainforth et al., 2023), wherein the cost of selecting an experiment that maximizes the information gain is negligible in comparison to the cost of obtaining the oracle answer.

by the LLM agent, it's outcome is the oracle answer, $a$, and the unknown quantity of interest, is the generated solution, $h$. Therefore, the information gain (IG) from obtaining the additional task specification as a pair $(q, a)$, given a problem statement, $\mathcal{S}$, can be formalized as:

$$\text{IG}(q, a) := \mathbb{H}[p^*(h|\mathcal{S})] - \mathbb{H}[p^*(h|\mathcal{S} \cup (q, a))], \tag{2}$$

where $\mathbb{H}$ is the Shannon entropy. However, because IG strictly depends on the answer $a$, which is unknown at the point of asking the question $q$, we need to consider the expected information gain–a quantity which arises by taking the expectation over all possible answers, given the query $q$:

$$\text{EIG}(q) := \mathbb{E}_{p^*(a|q,\mathcal{S})}[\text{IG}(q, a)] = \mathbb{H}[p^*(h|\mathcal{S})] - \mathbb{E}_{p^*(a|q,\mathcal{S})}\mathbb{H}[p^*(h|\mathcal{S} \cup (q, a)] \tag{3}$$

Noting that the first term of the EIG in (3) does not depend on $q$, we focus on the second term:

$$-\mathbb{E}_{p^*(a|q,\mathcal{S})}[\mathbb{H}[p^*(h|\mathcal{S} \cup (q, a))]]. \tag{4}$$

In the above, $p^*(a|q, \mathcal{S})$ denotes the distribution of oracle answers to a question $q$, given $\mathcal{S}$. We will assume that the law of total probability applies to $p^*$ so that $p^*(a|q, \mathcal{S}) = \sum_{h \in \mathcal{H}} p^*(a|q, h)p^*(h|\mathcal{S})$. We will also assume that $p^*$ is an oracle that given a sample solution $h$, can always answer a question $q$, about $h$, truthfully, so that $p^*(a|q, h) > 0 \Rightarrow h \vdash (a, q)$. Under this assumption, all plausible answers $a$ to a question $q$ must be semantically equivalent. Thus, without loss of generality, we can assume that for each question, $q$, there exists only one answer, $a$, describing $h$, for which $p^*(a|q, h) = 1$. Therefore, each question $q$, generates a partitioning of $\mathcal{H}$ into non-overlapping sets of solutions $\mathcal{H}_{(q,a)}$ compatible with $\mathcal{S} \cup (q, a)$. Let $\mathcal{A}_q$ denote the space of all unique answers to a question $q$ given the current problem statement $\mathcal{S}$ and let $\mathcal{H}_{(a,q)} := \{h \in \mathcal{H} : p^*(a|q, h) = 1\}$, then we must have that:

$$\mathcal{H} = \bigcup_{a \in \mathcal{A}_q} \mathcal{H}_{(q,a)} \quad \text{and} \quad \mathcal{H}_{(q,a)} \cap \mathcal{H}_{(q,a')} = \varnothing \quad \forall a, a' \in \mathcal{A}_q \quad \text{s.t.} \quad a \neq a'.$$

In this view, the utility of each question can be seen through the partitioning of $\mathcal{H}$ that it generates and the compound likelihood of solutions belonging to each of the partitions $\mathcal{H}_{(q,a)}$, $a \in \mathcal{A}_q$.[2]

**Requirement querying.** Note that the ground-truth information gain depends on the unknown distribution $p^*$. Ideally, we would like the generative distribution of the LLM, $p_{\phi_h}(\cdot|\mathcal{S})$, to closely approximate $p^*(\cdot|\mathcal{S})$ for any problem statement, $\mathcal{S}$. This is, however, not guaranteed. The bias of $p_{\phi_h}$ may not adequately represent the preferences of the problem setter encoded in $p^*$. However, if, both $p^*$ and $p_{\phi_h}$ can be decomposed as a product of the objective function $\mathbb{1}\{h \vdash \mathcal{R}\}$ with their subjective counterparts, then $\mathbb{1}\{h \vdash \mathcal{R}\}$ provides a source of common grounding between the two distributions. Our strategy of active task disambiguation will therefore solely focus on requirement querying. With that in mind, from now on we will assume that each question-answer pair, $(q, a)$, yields a new requirement, extending the current set of requirements and progressively reducing the set of compatible solutions (c.f. Fig. 2b).

**Uniformity of the unknown.** As $p^*$ is unknown and $p_{\phi_h}$ may be inadequately biased, our strategy of active task disambiguation will be agnostic to the biases of both $p^*$ and $p_{\phi_h}$. We will therefore approximate $p^*$ with a uniform distribution over the set of all solutions compatible with the given set of requirements. Under this assumption, we have that

$$\mathbb{E}_{p^*(a|q,\mathcal{S})}[\mathbb{H}[p^*(h|\mathcal{S} \cup (q, a))]] = \sum_{a \in \mathcal{A}_q} \mathbb{H}[p^*(h|\mathcal{S} \cup (q, a))] \sum_{h' \in \mathcal{H}} p^*(a|q, h')p^*(h'|\mathcal{S}) \tag{5}$$

$$= \sum_{a \in \mathcal{A}_q} \mathbb{H}[p^*(h|\mathcal{S} \cup (q, a))] \sum_{h' \in \mathcal{H}_{(q,a)}} \frac{1}{|\mathcal{H}|} \tag{6}$$

Given that $p^*(\cdot|\mathcal{S})$ is uniform on $\mathcal{H}$ we have that $p^*(h|\mathcal{S} \cup (q, a))$ is uniform on $\mathcal{H}_{(q,a)}$, thus $\mathbb{H}[p^*(h|\mathcal{S} \cup (q, a))] = \log(|\mathcal{H}_{(q,a)}|)$, and so

$$\mathbb{E}_{p^*(a|q,\mathcal{S})}[\mathbb{H}[p^*(h|\mathcal{S} \cup (q, a))]] = \frac{1}{|\mathcal{H}|} \sum_{a \in \mathcal{A}_q} |\mathcal{H}_{(q,a)}| \log(|\mathcal{H}_{(q,a)}|) \tag{7}$$

We note that the above is minimized when $\mathcal{H}_{(a,q)}$ are of equal size.

---

[2]In practice, when $p^*$ is represented through an internal likelihood function of a human, the two assumptions on the law of total probability and truthfulness of $p^*$ may be violated; humans may answer questions incorrectly or answer "I don't know", giving no information about $\mathcal{H}^*$. We see such events as unpredictable random noise that we decide not to model for simplicity. Alternative approaches may penalize questions $q$ for which the expected uncertainty of $p^*(a|q, h)$ is high.

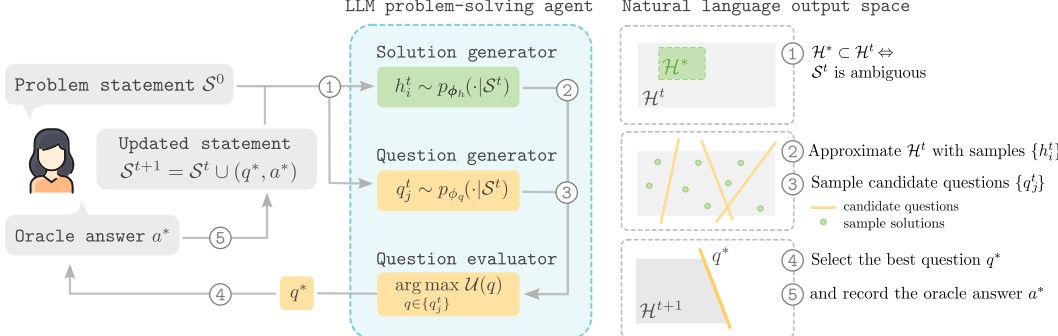

Figure 3: *Active task disambiguation.* ① Starting from $t = 0$, the problem statement $\mathcal{S}^t$ is presented to the problem-solving agent. ② The agent reasons about the problem to infer the set of solutions $\mathcal{H}^t$ compatible with the requirements of $\mathcal{S}^t$. In order to approximate $\mathcal{H}^t$, a set of candidate solutions $\{h_i^t\}$ is sampled. ③ To discern between different solution variants, the agent generates candidate questions $\{q_j^t\}$. ④ A question $q^*$ with the highest utility is selected and presented to the oracle. ⑤ Based on the oracle answer, $a^*$, the problem statement is extended by the new specification defined through $(q^*, a^*)$; the process can be repeated with the updated problem statement $\mathcal{S}^{t+1} = \mathcal{S}^t \cup (q^*, a^*)$ resulting in a reduced space of compatible solutions, $\mathcal{H}^{t+1} \subset \mathcal{H}^t$.

**Corollary 1.** *Let $\mathcal{H}$ be the set of solutions compatible with the requirements $\mathcal{R}$ within the problem statement $\mathcal{S}$. Suppose that $p^*(\cdot|\mathcal{S})$ is uniform on $\mathcal{H}$. Let $\mathcal{Q}_n$ be the set of all questions with exactly $n$ possible answers such that for any $q \in \mathcal{Q}_n$, there exists a finite set of possible answers $\mathcal{A}_q$ with $|\mathcal{A}_q| = n$ and that each question-answer pair, $(q, a)$, induces a new requirement. Then, the EIG is maximized for a question $q^*$ whose possible answers in $\mathcal{A}_{q^*}$ partition $\mathcal{H}$ into equally sized subsets.*

> 💡 If we only consider binary questions with $|\mathcal{A}_q| = 2$ (e.g. yes-or-no type of questions), then the question with the highest information gain is the one which partitions the set of all possible solutions $\mathcal{H}$ into two subsets of equal size. The second pane on the right of Figure 3 shows how at iteration $t$, three candidate questions may split the space of compatible solutions into subsets; the selected question $q^*$ is one which results in the most even partitioning.

> 💡 From (7) it follows that $\max\limits_{q \in \mathcal{Q}_n} \text{EIG}(q) \propto \log(\frac{|\mathcal{H}|}{n})$. This implies that a question with more possible answers can result in larger information gain. However, this requires finding a categorization of the possible solutions that partition them into balanced subsets. In the extreme case, if $|\mathcal{H}| = K$, the question with the highest information gain is the one whose answer always points to exactly one of the solutions $h \in \mathcal{H}$. Perhaps, the only way to ask such questions is to enumerate all solutions in $\mathcal{H}$ and ask: "*Is $h^*$ is supposed to be $h_1, \dots h_{K-1}$ or $h_K$*"? Arguably, this is neither a natural question to ask nor a user-friendly one, as it requires the user to examine each of the $K$ possible solutions. We argue that questions with a small set of possible answers, yet ones that induce a balanced partitioning of the solution space strike a good balance between information gain and the mental load from the user.

## 3 THE METHOD: ACTIVE TASK DISAMBIGUATION

The key point of the previous section is that given the unknown bias of $p^*$, the question with the largest information gain is the one that splits the space of compatible solutions into equal partitions. Generating such a question requires the LLM agent to reason about the set of possible solutions at each interaction step and identify the features that discern them. Such a reasoning process can be viewed as a form of meta-cognitive ability, requiring the LLM agent to consider the uncertainty within its own generative distribution. While a straightforward solution to asking such clarifying questions may be to simply perform zero-shot prompting of the LLM agent, we hypothesize that the out-of-the-box abilities of LLMs in this form of reasoning are lacking in comparison to their solution-generating abilities. This may be caused by a relatively small number of clarifying questions present in their pre-training corpus. Our proposed method explicitly evaluates the utility of candidate questions via a small sample of self-generated solutions and returns the question that maximizes this utility. Fig. 3 shows a high-level overview of the workflow.

Let $\mathcal{U} : \Sigma \to \mathbb{R}$ denote a utility function determining the usefulness of a candidate question $q$. Based on the derivations of the previous section we define $\mathcal{U}$ as:

$$\mathcal{U}(q) = \text{EIG}(q) - c(q),$$

where $c : \Sigma \to \mathbb{R}_+$ is a cost function determining the expected cost of obtaining the oracle answer to a candidate question $q$[3]. Our method of active task disambiguation consists of the following steps. At each iteration $t$,

1. Sample a set of $N$ candidate solutions $\{h_i^t\}_{i=1}^N \sim p_{\phi_h}(\cdot | \mathcal{S}^t)$;

2. Sample a set of $M$ candidate questions $\{q_j^t\}_{j=1}^M \sim p_{\phi_q}(\cdot | \mathcal{S}^t)$;

3. Obtain pseudo answers $a_{i,j}^t$ for each question $q_j^t$ about every solution $h_i^t$;

4. Approximate the utility $\mathcal{U}$ of each question $q_j^t$ based on the answers $\{a_{i,j}^t\}_{i=1}^N$;

5. Return the question $q^* = \arg\max_{q \in \{q_j^t\}} \mathcal{U}(q)$;

6. Record the user answer $a^*$ and extend the problem statement to $\mathcal{S}^{t+1} = \mathcal{S}^t \cup (q^*, a^*)$.

In the above, $p_{\phi_q}$ correspond to the question-generating distribution of the problem-solving LLM. Step 3. requires approximating the EIG of the candidate questions $\{q_j^t\}_{j=1}^M$ based on the sample $\{h_i^t\}_{i=1}^N$. This estimate is computed by generating to all questions $q_j^t$ answers $\{a_{i,j}^t\}$ about each sample solution $h_i^t$. Depending on the type of the solutions, these answers may be generated by the problem-solving LLM itself (see experiment 4.1), or evaluated with an external tool (see experiment 4.2). Algorithm 1 shows the general procedure of estimating the EIG score based on equation (7).

---

**Algorithm 1** $\texttt{estimate\_EIG}(q_j, \{a_{i,j}\}_{i=1}^N)$

**Require:** A question $q_j$ and a set of $N$ answers $\{a_{i,j}\}_{i=1}^N$
$\{a_1, \ldots, a_n\} \leftarrow$ Unique answers in $\{a_{i,j}\}_{i=1}^N$
**for** $k \in \{1, \ldots, n\}$ **do**
  $n_k \leftarrow |\{i : a_{i,j} = a_k, i \in [N]\}|$
  $p_k \leftarrow n_k / n$
**end for**
**return** $- \sum_{k=1}^n p_k \log(p_k)$

---

> We note that the estimation of the EIG score, and consequently the selection of questions, directly depends on the sample $\{h_i^t\}_{i=1}^N$. The proposed approach shifts the load of selecting the best question from implicit reasoning about the best question to ask to explicit reasoning in the solution space. The underlying motivation for this strategy is that generating discriminative questions without direct access to the set of compatible solutions is more challenging than solution generation itself. Question selection via direct maximization of the EIG bootstraps the question-generating skills of LLMs using their potentially stronger solution-generating skill. Based on this observation, we make a couple of practical remarks.

**Ensuring uniformity.** When estimating the EIG score based on samples from $p_{\phi_h}(\cdot | \mathcal{S} = (\mathcal{R}, \mathcal{C}))$ we want to ensure that $p_{\phi_h}$ is close to uniform on the set of $\mathcal{R}$-compatible solutions. In practice, to encourage diversity of samples we may adopt two strategies. 1) Generate sample solutions with increased sampling temperature. 2) Instruct the LLM to generate a list of *"diverse and representative"* solutions. We find that adding this statement to the solution-generating prompt improves the diversity of generated samples and the coverage of $\mathcal{H}$ (see study 2. of experiment 4.1).

**Mitigating LLM noise.** The effectiveness of questions selected by maximizing the estimated information gain relies on the agent's ability to accurately evaluate $\mathbb{1}\{h \vdash \mathcal{R}\}$. For this, we need to ensure that a) the generative distribution of the agent, $p_{\phi_h}(\cdot | \mathcal{S})$, is error-free, i.e. $p_{\phi_h}(h | \mathcal{S} = (\mathcal{R}, \mathcal{C})) = 0$ for any $h \nvdash \mathcal{R}$ and b) the pseudo answers $a_{i,j}$ used to estimate the EIG are also error-free, i.e. $h_i \vdash (q_j, a_{i,j})$. While in select cases ensuring the compatibility of solutions with requirements is straightforward (see code generation in experiment 4.2), in many problems in which the agent generates solutions without any external grounding, ensuring that points a) and b) are error-free is challenging due to LLM hallucinations (Zhang et al., 2023b; Rawte et al., 2023). We observe that in practice, employing more refined prompting strategies, like CoT or self-reflection, works well (see study 3. of experiment 4.1).

---

[3]In the experimental section, for simplicity, we take $c$ to be constant. In the case of human-written answers, alternatives may include penalties for the question's length or the expected length of the answer.

# 4 EXPERIMENTS

Our experiments are designed to investigate two hypotheses which result from the discussions contained in the previous sections:

**H1)** Implicit reasoning about solutions to generate the most effective clarifying question is a difficult skill for LLMs. This skill can be improved by shifting the reasoning load from the question space to the solution space.

**H2)** The gap between implicit reasoning, i.e. generating questions without explicitly sampling hypothetical solutions, and explicit reasoning through a sample of solutions to select the best question is most significant in cases where:

   **H2a)** The LLM can generate representative and diverse samples of solutions, uniformly covering the space of solutions compatible with the given problem statement.

   **H2b)** The evaluation noise of the EIG is minimal. This is particularly true in cases when evaluation can be offloaded to an external tool.

Given the above, we will present two kinds of problems: one in which there is no external evaluator guaranteeing that sample solutions adhere to the given requirements and one in which we can ground the evaluation of $\mathbb{1}\{h \vdash \mathcal{R}\}$ with an external tool. For both experiments, prompts used to generate questions, solutions, and answers are provided in the Appendix (D.1.1 and D.2).

## 4.1 YES-OR-NO QUESTIONS WITH THE 20 QUESTION GAME

The 20 Questions game is a classic guessing game that involves one player (A) thinking of an object, and the other player (B) asking up to 20 yes-or-no questions to guess what it is. The object can be anything, often categorized into an animal, a place, or a person to give the guessers a starting point. The goal for the guessers is to identify the object with as few questions as possible. Despite the 20 questions game being seemingly a toy example, it provides an ideal setup to evaluate the multi-turn questioning abilities of LLM agents. Moreover, it serves as a parallel to many real-world applications, like conversational search, content recommendation, or even medical diagnosis.

### 4.1.1 THE MAIN EXPERIMENT

**Setup.** To reduce the sampling costs, we play the game for 10 instead of the original 20 rounds. We restrict the game to the category of animals. Here, the set of acceptable solutions, $\mathcal{H}^*$ are singletons, $\{h^*\}$ where $h^*$ represents a single animal name that player A may think about. Player A is simulated with GPT-4o-mini prompted to answer questions about the ground-truth animal $h^*$. We emphasize that in this setup there is no pre-fixed list of candidate animals that the user or the reasoning agent can choose from. Instead, at each point of the interaction, the guessing LLM (Player B) is free to guess any animal across the entire animal kingdom. For a quantitative analysis of question-generating strategies, we run the game on 15 arbitrary tasks corresponding to 15 animal names (see appdx. D.1.2). For each task, we run the iterative requirement querying for 10 iterations across 5 seeds.

**Question generation.** We consider four alternative methods for generating questions:

- *implicit*: the agent is prompted to generate a single candidate question;

- *implicit-ToT* Yao et al. (2023): using the same prompt, we sample $M = 5$ questions and prompt the LLM to select the best questions among the set of self-generated candidate questions;

- *EIG-uniform*: using the same prompt, we sample $M = 5$ questions and select the one that maximizes the estimated EIG score. The EIG is estimated assuming uniformity of sample solutions;

- *EIG-logprobs*: same as above, but the EIG is estimated using the log-probabilities of the sample solutions as returned by the LLM agent.

**EIG estimation.** To estimate the EIG for the latter two strategies, at each interaction step, we prompt the LLM agent to generate a list of $N = 20$ animals that adhere to the current set of requirements $\mathcal{R}^t$. If the list of requirements $\mathcal{R}^t$ is long, the LLM may generate animals that do not fulfill all the requirements. To mitigate this, after the initial sampling of the animals, we loop over

all requirements $\mathcal{R}^t$ and use the self-critic answering prompt to check if all requirements $\mathcal{R}^t$ are satisfied. This extra filtering step is repeated twice (see Study 3. section 4.1.2 for ablation of this step). Animals that do not fulfill the requirements are rejected and sampling is repeated until $N$ animals are obtained. The same self-critic answering prompt is used to generate the pseudo answers $a_{i,j}$ to candidate questions $q_j$ about sample hypothesis $h_i$. The resulting set of answers is used to estimate the EIG according to Algorithm 1. Refer to Appendix C, Algorithm 4 for details on estimating the EIG using the log-probabilities of the sample solutions $\{h_i\}_{i=1}^N$.

**Evaluation.** Our metric of success is the likelihood of the problem-solving agent guessing the ground truth animal given the obtained set of requirements, $\mathcal{R}^t$, at each interaction step. We approximate this likelihood by sampling 10 animals with $p_{\phi_h}(\cdot|\mathcal{R}^t, \mathcal{C})$ and then prompting the agent to order the animals from most to least likely. We record the position of $h^*$ within the returned list, counting from the bottom–a score of 0 indicates that $h^*$ has not been sampled and a score of 10 indicates that $h^*$ is at the top of the list. For each task, seed, and iteration, we compute this score with 25 sampling seeds.

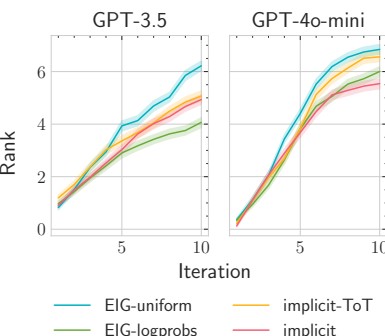

**Results.** As observed from Fig. 4, the *EIG-uniform* strategy outperforms the remaining strategies by a significant margin. The fact that it outperforms both *implicit* and *implicit-ToT* strategies confirms H1. Shifting the reasoning load from question space to direct reasoning about solutions results in improved efficacy of the selected questions–the elicited requirements lead to a higher likelihood of the LLM guessing the right animal after fewer iterations. We also observe that *EIG-logprobs* significantly underperforms in comparison to

Figure 4: *Comparison of question-generating strategies on the game of 20 questions.* Rankings averaged across 15 ground-truth animals, 5 run seeds, 25 evaluation seeds. See Appendix E.1 for results with the Llama family models.

*EIG-uniform.* This proves our claim about the LLM's generative distribution $p_{\phi_h}$ being inadequately biased, favoring solutions that do not necessarily agree with the ground truth solution set $\mathcal{H}^*$. Finally, we note that the gap in performance between *EIG-uniform* and the two implicit strategies is much lower for GPT-4o-mini, which is considered to be overall a significantly more capable model than GPT-3.5-turbo, especially when more reasoning steps are enforced with the ToT prompt.

### 4.1.2 ADDITIONAL STUDIES

The aim of this section is to gain further insights and investigate the impact of the design choices behind the EIG-uniform strategy on the effectiveness of the generated questions. In what follows, we present results for the problem-solving agent implemented with the GPT-3.5-turbo model.

**Study 1: Questions' utility.** While we do not restrict the game to a pre-fixed list of animals, in order to evaluate the partitioning properties of generated questions, we construct a large and diverse list of 500 animals (see Appendix D.1.2). Figure 5 shows the number of animals compatible with the requirements at the first six iterations from the large list constructed, instead of the small samples of $N$ animals used during the reasoning process (results limited to the first six rounds due to high costs of responding to all questions about 500 animals). We find that questions generated with the EIG-uniform strategy result in smaller subsets of $\mathcal{R}^t$-compatible

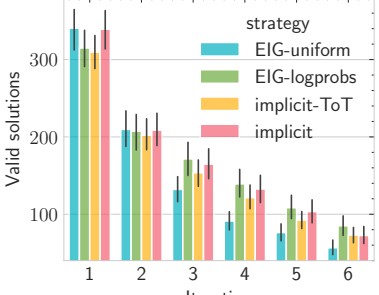

Figure 5: *Number of valid solutions after each iteration.*

solutions, and thus more effective disambiguation. This also shows that the estimation of EIG based on just $N = 20$ samples during question elicitation is a good proxy for the ground truth EIG, here approximated with the large list of 500 animals.

**Study 2: Diversity of solutions.** Estimation of questions' utility is dependent on the quality of sample solutions, which need to be sufficiently diverse to approximate $\mathcal{H}^t$ well. To ensure good coverage, the prompt for animal guessing includes the statement: *Generate a carefully selected, diverse, and representative set of animals.*. The EIG score used in our experiments is equivalent to the conditional mutual information (MI) between the random solutions $h \sim p_{\phi_h}(\cdot|\mathcal{S})$ and random answers $a \sim p_\phi(\cdot|q) := \sum_{h \in \mathcal{H}} p_{\phi_a}(\cdot|q, h) p_{\phi_h}(h|\mathcal{S})$, for a fixed question $q$. In order to test the impact

of the diversifying statement, we compute this score across a range of 330 diverse questions. From the main experimental results, we collect the unique set of 330 questions asked until the sixth iteration. We then run the game until the first six iterations sampling animals with and without the diversifying statement. We also vary the number of samples $N$. After obtaining the sets $\{h_i^t\}_{i=1}^N$ for all $t \in [1, \ldots, 6]$, we estimate the score for each of the 330 questions

Table 1: *Diversity of samples*

| $N$ | diverse | vanilla |
|---|---|---|
| 10 | **0.57** (0.01) | 0.46 (0.01) |
| 20 | **0.58** (0.01) | 0.54 (0.01) |
| 30 | **0.60** (0.01) | 0.58 (0.01) |

based on the LLM-generated answers. Assuming that the collection of 330 questions is diverse, the average of the conditional MI score will be maximized for $p_{\phi_h}$ close to uniform. Thus, a higher MI score implies greater diversity of the generated sample solutions. Table 1 confirms that the inclusion of the diversifying statement leads to more diverse samples of animals, validating H2a.

**Study 3: Requirement compatibility.** To mitigate the impact of LLM hallucinations on sampling solutions that do not adhere to the given set of requirements our solution generating prompt employs an extra filtering step to ensure that sample solutions $h_i$ indeed adhere to the current set of requirements. We test the impact of filtering on the accuracy of generated solutions. Figure 6 shows the average number of solutions rejected with a single filtering step (left) alongside the final rankings for the *EIG-uniform* strategy when 0, 1 or 2 filtering steps are used (right). As

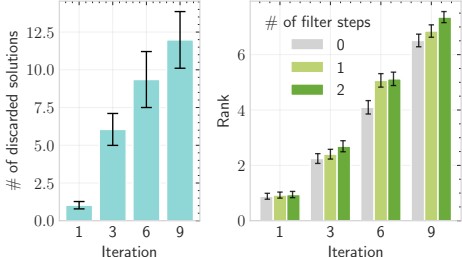

Figure 6: *The impact of LLM noise.*

the number of requirements increases, the LLM is more prone to hallucinations. A form of "self-reflection" via filtering becomes crucial in ensuring accurate outputs and confirming H2b.

## 4.2 ACTIVE CODE GENERATION–REQUIREMENTS AS UNIT TESTS

**Setup.** In our second experiment, we demonstrate active task disambiguation with an external tool ensuring a near error-free evaluation of requirement compatibility. In this experiment, the goal of our reasoning agent described in the initial prompt $\mathcal{S}^0$ is to generate a code solution $h$, based on the requirements $\mathcal{R}^0$ specified via a user-defined instruction describing the expected functionality of $h$. Following the setup of Chen et al. (2023), $\mathcal{S}^0$ contains a code snippet that includes statements such as imports, the function header, and a short comment describing the expected functionality of the generated code. Due to the ambiguous nature of natural language and the fact that at the point of writing the instruction not all edge cases might have been considered, $\mathcal{S}^0$ is likely to be ambiguous (see Appendix E.2.2 for examples).

**Question generation.** We consider two types of clarifying "questions":

**(B)** A question $q$ is a generated test case in the form of an assertion that the oracle is supposed to either confirm as correct or reject. We call these questions binary, as they only have two kinds of responses: True or False, similarly to the yes-or-no questions from the previous experiment.

**(O)** A question $q$ is a generated input to the desired function. The oracle returns the expected output of the code. We call these questions "open", as for one sample input there may exist a nearly unconstrained number of valid outputs, similarly to open-ended questions.

We compare questions generated zero-shot against questions selected by first sampling $M = 5$ candidate questions and then selecting one that maximizes the EIG, under the assumption of uniformity.

**Answers and requirements.** Both the ground truth answers $a^*$ and the answers $a_{i,j}$ used for EIG estimation are obtained by executing the ground-truth or a candidate program $h$, respectively, against a question $q$. The resulting question-answer pairs are turned into additional requirements as executable unit tests that each generated solution must pass and appended to $\mathcal{R}^t$. Generated programs are executed with an external Python interpreter in a sandbox environment. The "answering" of questions through an external tool ensures near noiseless estimation of the EIG score.

**Solution generation.** The solutions $h_i^t \sim p_{\phi_h}(\cdot|\mathcal{S}^t)$ are sampled by prompting the LLM to generate code completions which are then filtered to only those samples that pass the test cases in $\mathcal{R}^t$. This ensures that all solutions sampled from the LLM conform to the elicited requirements.

**Evaluation.** We evaluate all question-generating strategies on the HumanEval benchmark containing simple coding problems (Chen et al., 2021), and the more challenging APPS (Hendrycks et al.,

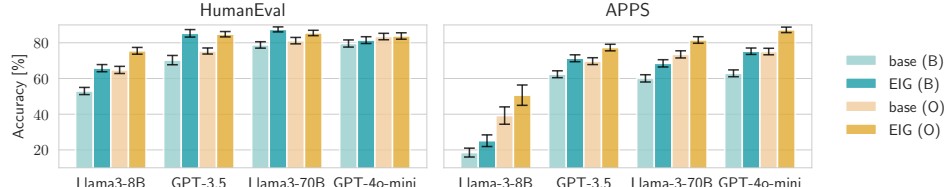

Figure 7: *Accuracy of generated code solutions after eliciting 4 additional requirements.* Results averaged across 48 tasks, 3 run and 3 evaluation seeds.

2021) benchmark with competition-level coding challenges[4], limiting the number of total questions asked to 4. After obtaining for each iteration $t$ a set of test cases $\mathcal{R}^t$, we evaluate the discriminative power of the obtained requirements, by sampling 20 solutions from $p_{\phi_h}(h|\mathcal{S}^t)$ and calculating the percentage of code solutions that pass the hidden test cases provided in the benchmark dataset.

**Results.** Figure 7 shows the accuracy of the generated code samples after eliciting 4 requirements with all strategies. Tables 3 and 4 in the appendix show the results across all 4 iterations. We observe that EIG-based strategies lead to higher accuracy of the outputs with fewer test cases queried. As expected by the theoretical discussion of section 2.1, questions with more possible answers (O) can be more informative than questions with only two possible answers (B). However, if these are to be answered by a human user, "open" questions arguably require an increased mental load to answer. We also observe that for more capable models, the gap between the baseline questioning strategies and their EIG-equivalent version is smaller. Results are consistent across both benchmarks, indicating that the EIG methods remain effective across varying levels of task ambiguity and difficulty[5].

We also note that the low accuracy of generated samples $h \sim p_{\phi_h}(\cdot|\mathcal{S}^0)$ does not imply that $\mathcal{S}^0$ itself is inherently ambiguous, as it may simply be caused by the LLM's limitations in following complex instructions. However, even if $\mathcal{S}^0$ is objectively not ambiguous, the input-output examples appended to the prompt $\mathcal{S}^t$ may positively bias $p_{\phi_h}$ towards correct solutions, which is a desirable by-product of employing active querying strategies to task disambiguation.

## 5 DISCUSSION

**Limitations.** While our work primarily focuses on efficient requirement elicitation, handling ambiguously specified tasks involves two equally important aspects determining a) that the given problem is ambiguous; b) when a sufficient number of requirements have been collected to stop querying the user. Kuhn et al. (2022) demonstrate that in select instances, ambiguity detection can be effectively resolved with zero-shot prompting. We believe that future research should explore alternative strategies. We also note that the question-generating strategies presented in this work require an increased number of LLM calls compared to the baselines (see Appendix F). However, in line with the assumptions commonly made in BED, we take the stance that the computational load required to select the optimal query is negligible compared to the value of acquiring information that reduces problem ambiguity. We anticipate this assumption will become more valid over time as technology advancements lower the costs of LLM token generation, thereby enhancing the importance of efficient information acquisition strategies.

**Conclusions and Impact.** Our findings suggest that clarifying questions generated with zero-shot prompting of LLMs are less efficient than those elicited by direct estimation of their utility with respect to the set of self-generated solutions. This suggests that the current skills of LLMs in generating efficient clarifying questions are underdeveloped, leaving room for improvement. Agents with well-developed meta-cognitive skills should be able to implicitly reason about the best question to ask without relying on multi-stage prompting strategies. We hypothesize that LLMs' deficiency in asking good clarifying questions stems from their limited exposure to such questions in the training corpus. To address this, our proposed framework offers a way to generate synthetic datasets of underspecified problems and their corresponding optimal clarifying questions. These datasets could serve as a resource for supervised fine-tuning, enhancing LLMs' abilities to disambiguate tasks more effectively and improving their interactive real-world problem-solving capabilities.

---

[4]We filtered the tasks of both benchmarks to a subset of 48 non-trivial tasks, i.e. tasks that do not achieve a near 100% zero-shot accuracy when sampling from $h \sim p_{\phi_h}(\cdot|\mathcal{S}^0)$ with GPT-3.5-turbo or GPT-4o-mini.

[5]The zero-shot accuracy of GPT-4o-mini on APPS is only $\sim$35% in comparison to the $\sim$70% achieved on HumanEval confirming the more challenging nature of the APPS benchmark.

## REPRODUCIBILITY

The pseudo algorithm to execute active question generation can be found in Appendix C. The exact format of the prompts used for both experiments is presented in Appendix D. Results are provided for two closed sourced models: GPT-3.5-turbo, GPT-4o-mini; and two source models: Llama-3-8B (Instruct), and Llama-3-70B (instruct). Code for reproducing the experimental results of section 4.2 is made available at: https://github.com/kasia-kobalczyk/active-task-disambiguation. The repository also includes generated programs and querries with GPT-3.5-turbo and GPT-4o-mini.

## ACKNOWLEDGMENTS

This work was supported by Azure sponsorship credits granted by Microsoft's AI for Good Research Lab. Katarzyna Kobalczyk is supported by funding from Eedi. Nicolás Astorga is sponsored by W.D. Armstrong Trust and Tennison Liu is sponsored by AstraZeneca. We thank the anonymous ICLR reviewers, members of the van der Schaar lab, and Andrew Rashbass for many insightful comments and suggestions.

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

# A EXTENDED RELATED WORK

**Active (in-context) learning.** In conventional AL, the ML system is designed to select queries from a fixed pool of unlabeled examples in order to reduce the uncertainty about its own outputs. Several works have explored AL strategies to improve the performance of LLMs on few-shot learning tasks performed with in-context learning Zhang et al. (2022b); Margatina et al. (2023); Diao et al. (2024). By acquiring new labeled examples, the generative distribution of the LLM is expected to shift towards outputs consistent with the ground-truth labels. In our setup, we do not have access to a fixed pool of questions that the agent can choose from. Instead, the agent generates a question on its own. Noting that a question can be of the form "What is the label $y$ for an input $x$?", active task elicitation can be seen as a generalization of AL. Furthermore, as observed by Zhang et al. (2022b), in-context learning performance can be highly unstable across sample examples due to the idiosyncrasies of how LLMs update their generative distribution when extending the set of in-context examples. Given the unpredictable nature of the LLMs' distribution, our work focuses on eliciting binary task requirements, enabling the agent to filter its own outputs that do not conform to task requirements specified by the user. By extending the problem statement with additional requirements, the bias of $p_{\phi_h}$ changes at each iteration, yet it remains unknown whether this change is aligned with $p^*$. By encouraging uniformity of $p_{\phi_h}$ over the set of compatible solutions, our active-reasoning framework steers the agent to consider many possible interpretations of a task at each point of the interaction, resulting in the selected questions being less biased towards most likely interpretations according to the possibly misaligned language model.

**Clarifying questions and generative task elicitation.** Before the emergence of LLMs, prior works (Rao & Daumé III, 2018; 2019; Min et al., 2020) have considered the problem of learning single-turn clarifying questions, with the question generator trained as sequence-to-sequence RNN's based on a pre-collected dataset of problems, clarifying questions, and their answers. More recently, in order to effectively address ambiguous user questions, Krasheninnikov et al. (2022) fine-tune the GPT-3 model on a data set of conversations consisting of ambiguous user requests, clarifying questions, and final answers. Kuhn et al. (2022) show that LLMs can reason about ambiguous aspects of a query and generate clarification questions with zero-shot prompting. Similarly, Li et al. (2025) capitalize on zero-shot prompting of LLMs and introduce a framework in which LLMs infer intended behaviour by querying the user with examples to label, yes-or-no questions or open-ended questions. They show that the LLM-generated queries are more efficient and require less effort than user-written prompts, enabling the discovery of initially unanticipated considerations of a task. Our work demonstrates that the LLM-generated questions can be improved by encouraging the agent to explicitly reason at inference time about the space of viable outputs given its current knowledge about the problem. This aligns with the principles highlighted in Groenendijk (1984), where reasoning about the semantics of questions plays a crucial role in shaping subsequent answers.

**Preference elicitation with LLMs.** A number of recent studies (Yang et al., 2021; Piriyakulkij et al., 2023; Handa et al., 2024; Austin et al., 2024) have leveraged LLMs for user preference elicitation, employing ideas of BED to select most informative queries. Despite surface-level similarities, these approaches are targeted at recommendation systems operating on a pre-determined set of objects or fixed feature spaces. For instance, (Handa et al., 2024) present an interactive preference elicitation framework wherein a linear Bayesian model is used to describe user preferences over a set of features elected prior to the start of user interaction. In this setup, the LLM's role is limited to feature extraction and query verbalization. In contrast, our paper focuses on scenarios where the LLM reasoning agent is expected to output a solution that belongs to an unconstrained space of natural language, not pre-determined by a feature space of fixed dimensionality nor a fixed list of hypothetical answers. This necessitates an iterative sampling of solutions at each interaction step to approximate the currently available options. Albeit more challenging, the unconstrained setting closely reflects the real-world usage of LLMs as general purpose reasoning agents. Moreover, it enables the LLM agent to query the user about aspects of hypothetical solutions at varying levels of granularity, and crucially, as noted by Li et al. (2025), ask about aspects of a given problem that could have been difficult to anticipate before engaging in the interactive dialogue.

**Ambiguity in natural language tasks.** Prior work highlights the importance of natural language ambiguity and its direct impact on LLM performance in instruction following. Liu et al. (2023) introduce a benchmark to assess LLMs' ability to recognize ambiguous sentences and disentangle their possible meanings. Tamkin et al. (2023) propose a benchmark of ambiguously specified classi-

fication tasks, requiring LLM agents to infer missing information. Niwa & Iso (2024), similar to our work, define ambiguity in terms of the sets of compatible and ground-truth solutions. Additionally, they introduce an ambiguity taxonomy that categorizes different types of instruction ambiguities and propose methods to refine initial instructions with clearer specifications.

**Other related works.** Recent studies have further explored the capabilities of LLMs in interactive and active learning scenarios. Hu et al. (2024) introduce Uncertainty of Thoughts (UoT), an algorithm designed to improve LLMs' information seeking by enabling them to ask effective questions. This approach uses uncertainty modeling to guide the LLM in actively reducing the LLMs uncertainty. Similarly, Astorga et al. (2024a) investigate active learning with LLMs in settings where data is partially observed and acquiring information has costs, focusing on efficient feature and label acquisition. Building upon the theme of question generation, Andukuri et al. (2024) propose STaR-GATE, a method that rewards language models for generating useful clarifying questions, enhancing their ability to resolve task ambiguity through self-improvement.

# B  POTENTIAL APPLICATIONS OF ACTIVE TASK DISAMBIGUATION

Active task disambiguation holds promise across a broad spectrum of applications. For instance, in personalized education and intelligent tutoring systems, clarifying questions can reveal subtle learning objectives and misconceptions, enabling real-time adaptation of instructional strategies (Aleven et al., 2016; Graesser et al., 2005; Krasheninnikov et al., 2022). In software engineering, interactive requirement elicitation helps mitigate misinterpretation during system design and development, leading to more efficient coding cycles and improved software quality (Jirotka & Goguen, 1994; Li et al., 2025). Creative industries—ranging from content generation and design to game development—may also benefit by aligning AI-generated outputs with users' evolving intensions (Amershi et al., 2019).

Beyond conventional machine learning tasks, active task disambiguation may be extended to a diverse array of problem-solving and decision-making domains. In scientific research, systematically clarifying experimental constraints can support more precise experiment design and hypothesis refinement (Montgomery, 2017). In robotics and autonomous systems, iteratively disambiguating mission goals and environmental constraints is key to achieving safer, more adaptive behavior (Paden et al., 2016; Thrun, 2002). In healthcare, actively refining diagnostic criteria and treatment protocols can improve patient outcomes by tailoring interventions to individual needs (Topol, 2019). In addition, we envision active task disambiguation to enhance areas like: autoformulation (Astorga et al., 2024b), informed machine learning and autoML (Kobalczyk & Schaar, 2025), legal reasoning (Rissland, 1988), financial advisory (De Prado, 2018), emergency response planning (Zhang et al., 2022a), urban planning (Ševčíková et al., 2007), policy-making support (Bowen & Zwi, 2005), context-aware testing (Rauba et al., 2024b), self-healing systems (Rauba et al., 2024a), or cybersecurity (Purushottam et al., 2024).

## C  ACTIVE TASK DISAMBIGUATION

---

**Algorithm 2** Active requirement elicitation

---

**Require:** Initial problem statement $\mathcal{S}^0 = (\mathcal{R}^0, \mathcal{C})$, Max number of iterations $T$, Number of solutions sampled per iteration $N$, Number of questions sampled $M$.
  **for** $t$ in $\{1, \ldots, T\}$: **do**
    $\{h_i^t\}_{i=1}^N \sim p_{\phi_h}(\cdot|\mathcal{S}^t)$
    $\{q_j^t\}_{j=1}^M \sim p_{\phi_q}(\cdot|\mathcal{S}^t)$
    **if** $M = 1$ **then**
      $q^* \leftarrow q_1^t$
    **else if** $M > 1$ **then**
      **for** $j$ in $1, \ldots, M$ **do**
        **for** $i$ in $1, \ldots, N$ **do**
          $a_{i,j}^t \sim p_{\phi_a}(\cdot|h_i^t, q_j^t)$
        **end for**
        $\text{EIG}\left[q_j^t\right] \leftarrow \texttt{estimate\_EIG}(q_j^t, \{a_{i,j}^t\}_{i=1}^N)$
      **end for**
      $q^* \leftarrow \arg\max_{q \in \{q_j^t\}_{j=1}^M} \text{EIG}\left[q\right]$
    **end if**
    $a^* \leftarrow \texttt{get\_oracle\_anser}(q^*)$
    $r^* \leftarrow \texttt{transform\_to\_requirement}(a^*, q^*)$
    $\mathcal{R}^{t+1} \leftarrow \mathcal{R}^t \cup \{r\}$
    $\mathcal{S}^{t+1} \leftarrow (\mathcal{R}^{t+1}, \mathcal{C})$
  **end for**

---

**Algorithm 3** $\texttt{estimate\_EIG}(q_j, \{a_{i,j}\}_{i=1}^N)$

---

**Require:** A question $q_j$ and a set of $N$ answers $\{a_{i,j}\}_{i=1}^N$
  $\{a_1, \ldots, a_n\} \leftarrow$ Unique answers in $\{a_{i,j}\}_{i=1}^N$
  **for** $k \in \{1, \ldots, n\}$ **do**
    $n_k \leftarrow |\{i : a_{i,j} = a_k, i \in [N]\}|$
    $p_k \leftarrow n_k / p_k$
  **end for**
  **return** $-\sum_{k=1}^n p_k \log(p_k)$

---

The methods of obtaining $p_{\phi_h}$, $p_{\phi_q}$, $p_{\phi_a}$ as well as $\texttt{transform\_to\_requirement}$ and $\texttt{get\_oracle\_answer}$ subroutines are application-specific. We provide the specific prompts used in sections D. The experiment of the 20Q game 4.1 also considers an alternative method for estimating the EIG based on the token log-probabilities of sample solutions $\{h_i\}_{i=1}^N$:

---

**Algorithm 4** $\texttt{estimate\_EIG\_logp}(q_j, \{a_{i,j}\}_{i=1}^N, \{p_i\}_{i=1}^N)$

---

**Require:** A question $q_j$, the set of $N$ answers $\{a_{i,j}\}_{i=1}^N$, log-probabilities of sample solutions $\{p_i\}_{i=1}^N$
  $\{a_1, \ldots, a_n\} \leftarrow$ Unique answers in $\{a_{i,j}\}_{i=1}^N$
  **for** $k \in \{1, \ldots, n\}$ **do**
    $p_k \leftarrow \sum_{\{i : a_{i,j} = a_k\}} \exp(p_i)$
  **end for**
  **for** $k \in \{1, \ldots, n\}$ **do**
    $p_k \leftarrow p_k / \sum_{r=1}^n p_r$
  **end for**
  **return** $-\sum_{k=1}^n p_k \log(p_k)$

---

# D    PROMPTS AND EXPERIMENTAL DETAILS

## D.1    THE 20 QUESTIONS GAME

### D.1.1    PROMPT TEMPLATES

When eliciting the requirements for the game of 20 questions we use the following set of prompts:

**Solution generating prompt**

```
As an AI assistant, your role is to generate a wide and diverse range
of animals that strictly meet the specified requirements.  Your
objective is to guess an animal from the entire animal kingdom that
satisfies these requirements:
    {List of requirements R^t}
For this, generate a carefully selected, diverse, and representative
set of {N} animals following this scheme:

        1.  <H>
        2.  <H>
        ...
        {N}.  <H>

Fill <H> with full name animals.
```

**Question generating prompt**

```
Your objective is to guess an animal from the entire animal kingdom
that satisfies the following requirements:
    {List of requirements R^t}
For this, generate the most informative yes/no question."
```

**Question selection prompt (used only for implicit-ToT)**

```
Your objective is to guess an animal from the entire animal kingdom
that satisfies the following requirements:
    {List of requirements R^t}
For this, select the most informative yes/no question from this list:"
    {List of questions {q_j^t}_{j=1}^M
```

**Answering prompt, $\{a_{i,j}^t\}_{i=1}^N \sim p_{\phi_a}(\cdot | \{h_i^t\}_{i=1}^N, q_j^t)$**

```
You are an expert critic that specializes in responding to yes/no
questions.  Given these animals:
    1.  {h_1^t}.   <A>.
    2.  {h_2^t}.   <A>.
    ...
    N. {h_N^t}.   <A>.
For each animal fill <A> with 'Yes' or 'No' with the response for this
question:
    {Sample question q_j^t}
```

**Requirement transform prompts**

**Question to statement:**
```
Transform the following question into an affirmative statement q
You should start with 'The animal'.  Do not write anything else than
the affirmative statement.
```

**Statement negation:**
```
Transform the following question into an affirmative statement
statement
You should start with 'The animal'.  Do not write anything else than
the affirmative statement.
```

---

**Oracle answer prompt (for GPT-4o-minisimulating player A)**

```
Given the animal: h* and the question: q*, respond to the question
depending on the given animal.  You can only respond one of the
following answers.

        Yes: If the answer is yes.
        No: If the answer is no.
        Pass: If the answer could be either yes or no.

Think step by step.  After your thinking process respond:  'The final
answer is <Response>'.
```

---

### D.1.2 EXPERIMENTAL DETAILS

**Animals for the main setup.** The focus of the main experiments is to evaluate the efficacy of the requirements elicited with various question-generating strategies. To do this, we select an arbitrary list of 15 animals as the ground-truth solutions $h^*$. These are: 'Chameleon', 'Ladybug', 'Swordfish', 'Armadillo', 'Crab', 'Grasshopper', 'African Grey Parrot', 'Manta Ray', 'Rabbit', 'Flamingo', 'Pangolin', 'Toucan', 'Starfish', 'Tarantula', 'Salamander'. For each $h^*$, we run the game for $T = 10$ iterations and with 5 different seeds. For solution and question generation we use $N = 20$ and $M = 5$, respectively.

**Construction of the list of 500 animals for study 1.** As discussed in the theoretical section of our paper, a question that splits the solution space into roughly equal parts should result in the highest information gain. Since we do not restrict the user nor the reasoning agent to a pre-fixed list of animals, we cannot directly assess the partitioning properties of a question. Instead, during active question generation we resolve to sampling of individual $h_i$'s which should approximate the entire solution space. The aim of this study is to evaluate the partitioning properties of the generated questions on a much larger sample of solutions. To this end, we construct a large and diverse list of 500 animals. This list was obtained by prompting GPT-4o-minito stratify the animal kingdom into several categories. In response we obtained the following categories and their cardinalities: 100 mammals, 100 birds, 100 invertebrates, 50 amphibians, 50 reptiles and 100 fish. Then, we subsample the respective number of animals in each category from the animals generated within our main experimental results until the sixth iteration. The stratification ensures that the constructed list is sufficiently diverse.

## D.2 CODE GENERATION

### D.2.1 PROMPT TEMPLATES

When eliciting the requirements for the game of 20 questions we use the following set of prompts:

---

**Code header for $\mathcal{S}^t$**

```
def <function_name>(<input_name>):
    """
    <Comment explaining the expected functionality of the code>

    Examples:
    <Generated test cases>
    """
```

---

**Solution generating prompt**

```
You are an expert Python programmer that specializes in coding tasks.
Complete the function given by the user.  Do not change the function
signature.  Do not add any additional commentary.  Do not import any
additional libraries.  Start your answer with the function signature.

<Code header for S^t>
```

---

**Question generating prompt (open)**

```
You are an expert Python programmer that specializes in solving
user-specified coding tasks.  To ensure you correctly understand user
specifications, you can query the user for expected program outputs
of sample inputs.  Given the function signature, generate <M> sample
inputs that will be most helpful in formalizing user intent.  Structure
your response as a list of function calls:

    1. <function_name>(input_1)
    2. <function_name>(input_2)
    ...
    <M>. <function_name>(input_<M>)

Do not generate any additional content beyond the numbered list of
function calls.  Do not repeat the same inputs as in the Examples
given.

<Code header for $\mathcal{S}^t$>
```

**Question generating prompt (binary)**

```
You are an expert Python programmer that specializes in solving
user-specified coding tasks.  To ensure you correctly understand
user specifications, you can write additional test cases.  Given the
function signature, generate <M> sample test cases that will be most
helpful in formalizing user intent.  Structure your response as a list
of assertion:

    1. assert <function_name>(input_1) == output_1
    2. assert <function_name>(input_2) == output_2
    ...
    <M>. assert <function_name>(input_<M>) == output_<M>

Do not generate any additional content or comments beyond the list of
assertions.

<Code header for $\mathcal{S}^t$>
```

### D.2.2 EXPERIMENTAL DETAILS

We conduct our experiments on two benchmakrs: HumanEval (Chen et al., 2023) and APPS (Hendrycks et al., 2021). To ensure a sufficient level of ambiguity of the problem statements, we filter the set of tasks in these benchmarks to the set of non-trivial tasks, i.e. task that cannot be solved with a 100% accuracy zero-shot, i.e. without acquiring additional requirements. In addition, the APPS benchmark is filtered to the set of tasks that do not contain any example input-output pairs in the original problem formulation. Resulting datasets contain 48 and 47 tasks, respectively.

## E  ADDITIONAL RESULTS

### E.1  THE GAME OF 20 QUESTIONS

Table 2 shows the final ranking results for the game of 20 questions.

### E.2  CODE GENERATION

### E.2.1  FULL RESULTS

Table 3 shows the accuracies of the generated code samples at each iteration on the HumanEval benchmark. Table 4 contains the corresponding results for the APPS benchmark. Despite the more challenging nature of the APPS benchmark, as indicated by the significantly lower accuracy across all models at $t = 0$, our conclusions still hold. The EIG-based strategies outperform their non-EIG equivalents by a significant margin, and the "open" type of questions result in larger information gains than "binary" questions.

Table 2: *Average ranks for the game of 20 questions.* Rankings averaged across 15 ground-truth animals, 5 run seeds, 25 evaluation seeds. Standard errors in brackets.

(a) GPT-3.5-turbo

| $t$ | EIG-uniform | EIG-logprobs | implicit-ToT | implicit |
|---|---|---|---|---|
| 1 | 0.8 (0.0) | 1.0 (0.1) | **1.2** (0.1) | 0.9 (0.0) |
| 2 | 1.5 (0.1) | 1.4 (0.1) | **1.7** (0.1) | 1.5 (0.1) |
| 3 | **2.4** (0.1) | 1.9 (0.1) | 2.4 (0.1) | 2.0 (0.1) |
| 4 | **2.9** (0.1) | 2.4 (0.1) | **3.0** (0.1) | 2.5 (0.1) |
| 5 | **3.9** (0.1) | 2.9 (0.1) | 3.4 (0.1) | 3.0 (0.1) |
| 6 | **4.1** (0.1) | 3.2 (0.1) | 3.7 (0.1) | 3.6 (0.1) |
| 7 | **4.7** (0.1) | 3.4 (0.1) | 4.0 (0.1) | 4.0 (0.1) |
| 8 | **5.0** (0.1) | 3.6 (0.1) | 4.5 (0.1) | 4.3 (0.1) |
| 9 | **5.9** (0.1) | 3.8 (0.1) | 4.8 (0.1) | 4.7 (0.1) |
| 10 | **6.2** (0.1) | 4.1 (0.1) | 5.1 (0.1) | 4.9 (0.1) |

(b) GPT-4o-mini

| $t$ | EIG-uniform | EIG-logprobs | implicit-ToT | implicit |
|---|---|---|---|---|
| 1 | **0.4** (0.0) | 0.3 (0.0) | 0.2 (0.0) | 0.1 (0.0) |
| 2 | **1.1** (0.1) | 0.9 (0.1) | **1.1** (0.1) | **1.1** (0.1) |
| 3 | **2.0** (0.1) | 1.7 (0.1) | **2.0** (0.1) | **2.1** (0.1) |
| 4 | **3.4** (0.1) | 2.6 (0.1) | 2.7 (0.1) | 2.9 (0.1) |
| 5 | **4.4** (0.1) | 3.8 (0.1) | 3.8 (0.1) | 3.7 (0.1) |
| 6 | **5.5** (0.1) | 4.7 (0.1) | 5.1 (0.1) | 4.5 (0.1) |
| 7 | **6.2** (0.1) | 5.1 (0.1) | 5.7 (0.1) | 5.1 (0.1) |
| 8 | **6.5** (0.1) | 5.5 (0.1) | 6.1 (0.1) | 5.3 (0.1) |
| 9 | **6.7** (0.1) | 5.7 (0.1) | 6.5 (0.1) | 5.4 (0.1) |
| 10 | **6.8** (0.1) | 6.0 (0.1) | 6.6 (0.1) | 5.5 (0.1) |

(c) Llama3-70B (Instruct)

| $t$ | EIG-uniform | EIG-logprobs | implicit-ToT | implicit |
|---|---|---|---|---|
| 1 | **1.3** (0.1) | 0.4 (0.0) | 0.8 (0.0) | 0.5 (0.0) |
| 2 | **1.5** (0.1) | 0.5 (0.0) | 1.2 (0.1) | 0.9 (0.1) |
| 3 | 1.7 (0.1) | 0.9 (0.1) | **2.0** (0.1) | 1.4 (0.1) |
| 4 | **2.4** (0.1) | 1.2 (0.1) | **2.4** (0.1) | 2.1 (0.1) |
| 5 | **3.0** (0.1) | 1.5 (0.1) | 2.8 (0.1) | 2.7 (0.1) |
| 6 | **3.8** (0.1) | 1.8 (0.1) | 3.1 (0.1) | 3.4 (0.1) |
| 7 | **4.3** (0.1) | 2.2 (0.1) | 3.7 (0.1) | 3.8 (0.1) |
| 8 | **5.1** (0.1) | 2.7 (0.1) | 4.1 (0.1) | 4.3 (0.1) |
| 9 | **5.4** (0.1) | 3.2 (0.1) | 4.6 (0.1) | 5.3 (0.1) |
| 10 | **6.3** (0.1) | 3.9 (0.1) | 5.1 (0.1) | 5.9 (0.1) |

Table 3: Accuracy (%) of solutions on HumanEval benchmark. (B)-solutions generated with "binary" questions. (O)-solutions generated with "open" questions. Results averaged across 48 tasks, 3 run and 3 evaluation seeds per task.

(a) GPT-3.5-turbo

| $t$ | base (B) | EIG (B) | base (O) | EIG (O) |
|---|---|---|---|---|
| 0 | 44.1 (1.9) | 44.6 (1.8) | 47.1 (1.4) | 47.0 (1.4) |
| 1 | 55.3 (2.5) | 66.8 (2.5) | 67.5 (1.7) | **74.4** (1.6) |
| 2 | 65.2 (2.5) | 78.4 (2.1) | 71.6 (1.7) | **81.4** (1.6) |
| 3 | 70.7 (2.6) | 82.2 (2.1) | 75.5 (1.7) | **84.5** (1.5) |
| 4 | 70.8 (2.6) | **85.6** (2.0) | 75.9 (1.7) | **85.0** (1.5) |

(b) GPT-4o-mini

| $t$ | base (B) | EIG (B) | base (O) | EIG (O) |
|---|---|---|---|---|
| 0 | 71.0 (2.1) | 69.6 (2.1) | 71.4 (2.1) | 70.6 (2.1) |
| 1 | 74.6 (2.0) | 73.6 (2.1) | 77.1 (2.0) | 79.5 (1.8) |
| 2 | 76.7 (2.0) | 78.9 (2.0) | 78.8 (1.9) | 83.9 (1.7) |
| 3 | 77.0 (2.0) | 80.6 (1.9) | 80.8 (1.9) | 83.7 (1.8) |
| 4 | 79.6 (2.0) | 81.5 (1.9) | 83.5 (1.8) | 83.8 (1.8) |

(c) Llama3-70B (Instruct)

| $t$ | base (B) | EIG (B) | base (O) | EIG (O) |
|---|---|---|---|---|
| 0 | 63.9 (2.0) | 63.8 (2.0) | 63.6 (2.0) | 63.6 (2.0) |
| 1 | 71.3 (2.0) | **79.3** (1.7) | 72.9 (2.0) | **79.7** (1.7) |
| 2 | 73.0 (2.0) | **82.5** (1.6) | 75.2 (2.0) | **82.8** (1.7) |
| 3 | 75.4 (1.9) | **84.7** (1.6) | 78.3 (1.9) | 84.2 (1.7) |
| 4 | 78.8 (1.8) | **87.5** (1.4) | 81.2 (1.8) | 85.5 (1.5) |

(d) Llama3-8B (Instruct)

| $t$ | base (B) | EIG (B) | base (O) | EIG (O) |
|---|---|---|---|---|
| 0 | 34.8 (1.7) | 34.7 (1.7) | 34.6 (1.7) | 34.7 (1.7) |
| 1 | 40.9 (1.9) | 48.8 (1.9) | 53.7 (1.9) | **62.3** (2.0) |
| 2 | 45.6 (1.9) | 56.2 (2.0) | 58.9 (2.0) | **70.5** (2.0) |
| 3 | 49.6 (2.0) | 60.4 (2.0) | 60.6 (2.1) | **74.4** (2.0) |
| 4 | 53.0 (2.0) | 65.8 (2.0) | 64.8 (2.0) | **75.5** (1.9) |

E.2.2 AMBIGUITY IN HUMANEVAL

We note that the low accuracy of generated samples $h \sim p_{\phi_h}(\cdot|\mathcal{S}^0)$ does not imply that $\mathcal{S}^0$ itself is inherently ambiguous, as it may simply be caused by the LLM's limitations in following complex instructions. However, even if $\mathcal{S}^0$ is objectively not ambiguous, the input-output examples appended

Table 4: Accuracy (%) of solutions on the APPS benchmark. (B)-solutions generated with "binary" questions. (O)-solutions generated with "open" questions. Results averaged across 47 tasks, 3 run and 3 evaluation seeds per task.

(a) GPT-3.5-turbo

| $t$ | base (B) | EIG (B) | base (O) | EIG (O) |
|---|---|---|---|---|
| 0 | 32.1 (1.6) | 31.2 (1.6) | 32.1 (1.6) | 32.9 (1.7) |
| 1 | 49.0 (1.9) | 49.5 (2.0) | 51.3 (1.9) | **58.4** (2.0) |
| 2 | 54.3 (1.9) | 61.0 (2.0) | 59.0 (1.9) | **68.3** (2.0) |
| 3 | 60.8 (2.0) | 67.5 (1.9) | 68.2 (1.8) | **74.6** (1.8) |
| 4 | 62.4 (1.9) | 71.3 (1.9) | 69.7 (1.9) | **77.3** (1.8) |

(b) GPT-4o-mini

| $t$ | base (B) | EIG (B) | base (O) | EIG (O) |
|---|---|---|---|---|
| 0 | 35.8 (1.1) | 35.9 (1.2) | 34.6 (1.1) | 35.9 (1.2) |
| 1 | 53.9 (1.7) | 62.0 (1.8) | 59.2 (1.7) | **68.0** (1.9) |
| 2 | 57.5 (1.8) | 67.7 (1.9) | 68.8 (1.7) | **83.0** (1.6) |
| 3 | 60.8 (1.9) | 72.9 (1.8) | 72.7 (1.8) | **85.7** (1.5) |
| 4 | 62.9 (1.9) | 75.3 (1.8) | 75.1 (1.8) | **87.2** (1.5) |

(c) Llama3-70B (Instruct)

| $t$ | base (B) | EIG (B) | base (O) | EIG (O) |
|---|---|---|---|---|
| 0 | 38.0 (1.9) | 38.3 (1.8) | 38.4 (1.8) | 37.8 (1.8) |
| 1 | 49.0 (2.0) | 54.7 (1.9) | 57.1 (2.0) | **63.4** (2.0) |
| 2 | 51.6 (2.0) | 60.1 (2.0) | 61.9 (2.1) | **74.9** (1.9) |
| 3 | 57.7 (2.0) | 65.7 (2.1) | 68.8 (2.0) | **78.3** (1.9) |
| 4 | 60.1 (2.0) | 68.5 (2.0) | 73.5 (2.0) | **81.6** (1.8) |

(d) Llama3-8B (Instruct)

| $t$ | base (B) | EIG (B) | base (O) | EIG (O) |
|---|---|---|---|---|
| 0 | 10.6 (1.5) | 8.5 (1.6) | 9.1 (1.6) | 9.7 (1.8) |
| 1 | 13.5 (1.9) | 17.0 (2.4) | **21.8** (3.0) | **25.3** (3.4) |
| 2 | 15.4 (2.1) | 19.7 (2.9) | **26.9** (3.8) | **28.4** (4.3) |
| 3 | 14.5 (2.1) | 22.7 (3.3) | **32.1** (4.6) | **34.2** (5.4) |
| 4 | 18.5 (2.5) | 26.9 (3.7) | 39.2 (4.9) | **50.7** (5.7) |

to the prompt $\mathcal{S}^t$ may positively bias $p_{\phi_h}$ towards correct solutions, which is a desirable by-product of employing active querying strategies to task disambiguation. To ensure, however, that the code-generation experiment is indeed appropriate to test our strategy for ambiguous prompts, we have listed in table 5 sample tasks with explanation of their ambiguity.

Table 5: Sample ambiguous problem statements from the HumanEval benchmark.

| problem statement | ambiguity |
|---|---|
| ```from typing import List

def separate_paren_groups(paren_string: str) -> List[str]:
    """ Input to this function is a string containing multiple
    groups of nested parentheses. Your goal is to
    separate those group into separate strings and return the
    list of those.
    Separate groups are balanced (each open brace is properly
    closed) and not nested within each other
    Ignore any spaces in the input string.
    """``` | How to handle characters in the input string other than parentheses and spaces? How to handle strings with non-balanced groups of parentheses? |
| ```from typing import List

def remove_duplicates(numbers: List[int]) -> List[int]:
    """ From a list of integers, remove all elements that occur
    more than once.
    Keep order of elements left the same as in the input.
    """``` | Should all duplicated elements be removed or one unique element retained?, e.g. `remove_duplicates([4, 4, 3, 5, 5]) ->[4, 3, 5]` or `[3]` |
| ```def is_bored(S):
    """
    You'll be given a string of words, and your task is to count
     the number
    of boredoms. A boredom is a sentence that starts with the
     word "I".
    Sentences are delimited by '.', '?' or '!'.
    """``` | Is a sentence starting with "I'm" a boredom or not? |
| ```def histogram(test):
    """Given a string representing a space separated lowercase
     letters, return a dictionary
    of the letter with the most repetition and containing the
     corresponding count.
    If several letters have the same occurrence, return all of
     them.
    """``` | How to handle letters that are not separated by spaces? |
| ```def cycpattern_check(a , b):
    """You are given 2 words. You need to return True if the
     second word or any of its rotations is a substring in the
     first word
    """``` | What is understood by a string rotation? |
| ```def f(n):
    """ Implement the function f that takes n as a parameter,
    and returns a list of size n, such that the value of the
     element at index i is the factorial of i if i is even
    or the sum of numbers from 1 to i otherwise.
    i starts from 1.
    the factorial of i is the multiplication of the numbers from
     1 to i (1 * 2 * ... * i).
    """``` | Should the sum of 1 to $i$ include $i$? |

# F  Costs of question elicitation

We note that the EIG-based question-generating strategies presented in this work require an increased number of LLM calls compared to the zero-shot baselines. In line with the assumptions commonly made in BED, we take the stance that the computational load required to select the optimal query is negligible compared to the value of acquiring information that reduces problem ambiguity. We anticipate this assumption will become more valid over time as technology advancements lower the costs of LLM token generation, thereby enhancing the importance of efficient information acquisition strategies. However, given that in many real-world applications design choices may be constrained by the LLM sampling costs, we include a comparison of the effective number of LLM calls required at each interaction step for the key strategies considered. In the below, $N$ is the number of generated solutions at each step, and $M$ is the number of generated candidate questions.

## F.1  20 Questions

| Strategy | Cost |
|---|---|
| implicit (baseline) | $O(1)$ |
| implicit-ToT | $O(M)$ |
| EIG | $O(N + M + NM)$ |

In the baseline strategy, only one call is needed to sample a single question, $q^*$.

In the implicit-ToT strategy, additional $M$ calls are needed to generate a set of $M$ candidate questions, from which one is selected.

In the EIG-based strategies (EIG-uniform and EIG-logprobs), $M$ calls are needed to sample a set of candidate questions $\{q_j\}_{j=1}^M$ and $N$ calls to sample the hypothetical solutions $\{h_i\}_{i=1}^N$. To select the best question, we need to estimate the EIG, which requires the LLM to answer each question, $q_j$, about every sampled solution, $h_i$, thus requiring additional $NM$ LLM calls.

## F.2  Code Generation

| Strategy | Cost |
|---|---|
| base | $O(1)$ |
| EIG | $O(N + M)$ |

In the code generation example, the evaluation of the EIG does not require additional LLM calls. The answers, $a_{i,j}$, are obtained by executing the code solution, $h_i$, against each question $q_j$.

