# OpenReview forum: "Active Task Disambiguation with LLMs"
_ICLR.cc/2025/Conference — ICLR 2025 Spotlight_

### Official Review · Reviewer_HcvV · 2024-10-31

**Soundness:** 2
**Presentation:** 2
**Contribution:** 2
**Rating:** 6
**Confidence:** 3

**Summary:**

The paper addresses the challenge of large language models (LLMs) dealing with ambiguously specified problems, which is common in real-world interactions. The authors propose a Bayesian Experimental Design framework to tackle task disambiguation by having LLM agents ask clarifying questions to acquire additional task specifications. This approach requires meta-cognitive reasoning, which the authors argue LLMs currently lack. They introduce a method to generate targeted questions that maximize information gain, shifting the reasoning from implicit to explicit. Empirical results show this method leads to more effective task disambiguation compared to question-space reasoning approaches. The paper contributes by identifying the need for new reasoning methods, formalizing task ambiguity, proposing a BED-based strategy for question generation, and evaluating its effectiveness in interactive task elicitation scenarios.

**Strengths:**

* It introduces a novel method to handle ambiguity in task specifications by leveraging the principles of Bayesian Experimental Design, which is a significant advancement in the field of LLMs.
* The paper provides a clear and formal definition of task ambiguity, which is crucial for framing and addressing the problem within a mathematical and computational context.

**Weaknesses:**

* Consider adding a concrete example of task ambiguity in the main text, rather than relegating it to supplementary materials. A practical example would help readers grasp this key concept more easily.
* Why just show the rank in Figure 3?

**Questions:**

* IG in Eq. (2) denotes information gain?
* What does $c(q)$ mean ? Does it mean cost of API usage (tokens) ? Please clarify it.

---

> ### Author Response · Authors · 2024-11-19
> **Response to your review**
>
> Thank you for taking the time to review our paper. Below we address the questions and concerned raised.
>
> ---
>
> **Example of task ambiguity in the main text.** Thank you for this excellent suggestion. We agree that a small example at the beginning of the paper would help the readers grasp the key concepts discussed more easily.
> **UPDATE:** We included a concrete example of an ambiguous problem statement as a continuation of Example 1 in the context of code generation.
>
> **Figure 3 ranks.** The measure of efficacy of a set of collected requirements until iteration $t$ should be the likelihood of the problem-solving agent (here an LLM) in generating the ground-truth solution $h^*$ given the set of requirements $\mathcal{R}^t$. In the context of natural language responses, this metric is non-trivial to estimate empirically due to the inherent randomness of LLM generations. In such contexts, it is standard to use the pass@k score, which indicates the accuracy of the LLM-generated responses, if it the LLM is allowed to make up to $k$ distinct guesses.  If we were to evaluate only the first, most likely response generated by the model, (this is equivalent to the pass@1 metric commonly used in the code-generation contexts), only very few runs would result in a non-zero accuracy, requiring potentially far more iterations to observe statistically significant results. Instead of providing the value of the pass@k metric for an arbitrarily fixed $k$, we instead provide the ranking of a candidate solution $h^*$ in the list of 10 candidate solutions ordered from least to most likely, which we believe is an easy to interpret score and can also be used to approximate the pass@k metric. I.e. if for the given run, the rank of $h^*$ is greater than $10-k$, it would contribute positively to the computation of the pass@k metric.
>
> **IG in Eq. (2).** Yes, the IG in equation (2) denotes the information gain. Thank you for noticing  that this abbreviation has not been properly introduced in the text. We have now addressed this and updated the manuscript accordingly.
>
> **The meaning of c(q).** We denote by $c(q)$ the cost of obtaining the oracle answer if we were to ask the question $q$. In our presentation we keep this abstract to allow for multiple interpretations depending on the specific application contexts. This cost can stand for the costs associated with obtaining an answer from the user in a text format (e.g. the expected length of their response), or the expected cost of running a computational experiment (e.g. number of FLOPs, GPU usage, etc.). In our experiments, for simplicity, we set $c(q)$ to a constant value, assuming that all candidate questions are equally difficult to answer by the oracle.
>
> We stress that this cost, however, is not the cost of generating the question or evaluating its utility. At the point of utility estimation, all of the candidate questions must have been already generated, and the associated costs incurred. In this work, we follow the principles of Bayesian Experimental Design (Rainforth et al.), wherein the costs associated with eliciting a good questions are marginal in comparison to the costs associated with providing an answer to this question. Consequently, the goal of our framework is to find a question that maximises the expected information gain, while minimising the costs of obtaining the answer from the oracle.
>
> **UPDATE:** Thank you for bringing this point to our attention. We have provided additional clarifications on the meaning of $c(q)$ in the revised version of our manuscript. Additionally, we have included a short discussion on the computaitonal costs associated with the different question-elicitation strategies discussed in this work in Appendix E of the updated manuscript.
>
> Reference
>
> Rainforth, Tom, Foster, Adam, Ivanova, Desi R., & Bickford Smith, Freddie. "Modern Bayesian experimental design." Statistical Science, vol. 39, no. 1, 2024, pp. 100–114. Institute of Mathematical Statistics.
>
> ---
>
> We appreciate the time and effort you’ve dedicated to reviewing our paper. We hope that the our answers and proposed changes to the manuscript address your concerns satisfactorily.
>
> Thank you again for helping us improve the quality of this work.
>
> The authors of submission #11335

---

> > ### Comment · Reviewer_HcvV · 2024-11-24
> > **Thanks for your response**
> >
> > Thanks for your response. My concerns have been addressed. I decide to increase my score from 5 to 6.

---

> > > ### Author Response · Authors · 2024-11-25
> > > **Thank you**
> > >
> > > Dear reviewer,
> > >
> > > We sincerely thank the reviewer for taking the time to carefully consider our responses. We are grateful for the updated score and appreciate your acknowledgment of the contributions and improvements we made.
> > >
> > > Thank you for your continued engagement in the rebuttal process.
> > >
> > > Kind regards,
> > >
> > > The authors of submission #11335

---

### Official Review · Reviewer_qfUE · 2024-11-03

**Soundness:** 4
**Presentation:** 4
**Contribution:** 3
**Rating:** 8
**Confidence:** 5

**Summary:**

This paper addresses task ambiguity, where the intended goal of the user is not clear/is underspecified. The proposed method seeks to reduce ambiguity by asking follow-up questions to the user that clarify their intent. The authors use LLMs to both perform tasks and generate questions, and implement an information-gain-based ranking method to obtain questions that best reduce uncertainty; the method is based on Bayesian Experiment Design, and tries to partition the space of possible solutions s.t. the partitions are maximally balanced. The method is tested on two tasks and three LLMs. First, the authors test on a 20-questions style game where the agent has to correctly guess an animal by asking questions about its properties. Here, their method results in a better ranking of the correct animal, and they show that their questions eliminate more possible alternatives than a baseline without information gain. The second domain is coding, where the authors show that their method is able to generate unit tests (which act as questions). Here, their unit tests generally result in improvements on HumanEval, both when unit tests are posed as true/false statements and when open-ended unit tests are permitted.

**Strengths:**

**Quality:** The work rigorously defines its problem and then offers a potential solution. The authors do a good job of illustrating that their method works/how it works on 20 questions, while still showing real-world results on a naturalistic dataset in a coding domain, where many users actually use LLMs on a daily basis. This is a major strength of the paper. The experiments themselves are generally clearly described and well-documented/well-executed. The authors also run across a large number of seeds and release code, making their approach more reproducible.

**Significance:** Ambiguity and task ambiguity are increasingly important problems as more people use LLMs and AI systems. The authors make a strong case in the introduction for why ambiguity is important and the kinds of risks posed. The clarification-based approach they suggest has the potential to improve AI safety by reducing misunderstandings, and takes a human-centric approach to dealing with ambiguity.

**Originality:** The information-gain approach to dealing with ambiguity that is proposed here is novel, and the definition of task ambiguity in definition 1 is valuable.

**Clarity:** The paper is generally well-written and clear. I appreciated the colored boxes that provide additional detail/clarification. I found the formalizations made the paper more precise and were accessible. Several questions I had were preemptively addressed, e.g. on L509-511 where the authors point out that adding unit tests could provide other avenues for improvement beyond ambiguity. The figures (especially figure 2) nicely illustrate the problem and method.

**Weaknesses:**

**Empirical gains vs. cost**: Looking at Fig. 3, it seems like the proposed method improves the rank by ~2 ranks at iteration 10 for GPT3.5 and ~1 for GPT-4o-mini. While these results are significant, I do wonder (especially given the smaller gain on GPT4o-mini) whether they track with the added computational cost of the method. It would be worth noting the cost (i.e. number of tokens, number of model calls) to be able to directly compare these methods, since some of the baselines like ToT also incur more computational cost.

**Baseline clarity**: It's not completely clear what the ToT baseline is. Is this tree-of-thought? If so, it should be cited.

**Questions:**

Relevant references:
- https://arxiv.org/abs/1805.04655
- https://stokhof.org/wp-content/uploads/2020/09/groenendijk-stokhof_ssqpa.pdf
- https://arxiv.org/abs/2004.10645
- https://arxiv.org/abs/2211.07516

---

> ### Author Response · Authors · 2024-11-19
> **Response to your review**
>
> Thank you for your detailed and thoughtful feedback. We greatly appreciate your positive assessment of our work's quality, significance, originality, and clarity. Below, we address the weaknesses and questions you raised and clarify points that may have been unclear.
>
> ---
>
> **Empirical gains vs. cost.** Thank you for bringing up the point on computational costs vs. empirical gains. **UPDATE**: To address this part, we have included a new section in the Appendix of the manuscript (Appendix E), where we provide a detailed comparison of the LLM sampling costs associated with the competing question generating strategies. Indeed, the EIG-based strategies incur a much higher costs than other approaches. However, in the context of BED (Rainforth et al.), this trade-off is often justified. Obtaining the result of an experiment (here the answer to a question) is typically assumed to incur a much higher cost than the computational effort required to select the experiment (or question) itself.
>
> Reference:
>
> Rainforth, Tom, Foster, Adam, Ivanova, Desi R., & Bickford Smith, Freddie. "Modern Bayesian experimental design." Statistical Science, vol. 39, no. 1, 2024, pp. 100–114. Institute of Mathematical Statistics.
>
> **Baseline clarity.**  Indeed, we refer to the ToT baseline as a variation of a tree-of-thought, with “trees” of depth of one, where all questions are generated by the LLM and subsequently selected based on the LLM's own judgement. We have now included the corresponding citation.
>
> **Additional references.** Thank you for highlighting these references. We have incorporated them throughout the manuscript and in the Related Work section of the updated version.
>
> ---
>
> Thank you again for your detailed and valuable feedback. We have made the relevant updates to our manuscript which we hope improve its overall clarity and presentation clarity.
>
> Kind regards,
> The authors of submission #11335

---

### Official Review · Reviewer_xMi4 · 2024-11-04

**Soundness:** 3
**Presentation:** 3
**Contribution:** 3
**Rating:** 8
**Confidence:** 3

**Summary:**

This paper studies the question/user intent disambiguation problem for LLM agents. When the initial query do not provide enough information for LLM agent to solve the task, the agent should actively ask user questions to better clarify the task to solve. This paper proposed a method that can select the best clarification question at every turn. In particular, the method first samples a set of possible answers based on the existing information, and a set of possible clarification questions, then the question is selected based on the EIG score (expected information gain). The question along with its answer is added to the problem context for the next round of prompting. This execution loop can repeat for a few iterations until reaching a final answer. The authors evaluated the proposed method on two datasets, 1) 20-question games about animal names and 2) HumanEval coding tasks. The results suggest that the propose method of selecting clarification questions leads to better ovrall results than vanilla zero-shot question generation.

**Strengths:**

1. The formulation of EIG scores for selecting the best clarification question is well designed.  Intuitively, the question that brings maximum information gain will narrow down the possible search space the most,  but how to prompt the model to generate the most informative question is challenging. The authors did a nice work by shifting the question generation problem to a question selection problem, which leverages LLM’s reasoning ability and potentially external tools that provides more accurate signals for the selection.

2. The proposed method of generating/selecting best clarification questions can have wide application to different tasks. For example, in many real-world applications where LLM agents can be deployed, e.g. travel agent or coding agent, the user queries can be ambiguous and miss critical information. The proposed method can potentially be applied to help the agent understand uesr intent in fewer turns and thus complete tasks more efficiently.

**Weaknesses:**

The benchmarks used in this paper are too simple. For example the 20-question games only require guessing animal names, and it is designed for asking clarification questions. For HumanEval, it also contains mostly simple coding problems. It would be better if we can see results from more challenging agent benchmarks such as SWE-bench.

**Questions:**

1. For the two benchmarks tested in this paper, the EIG scores can be easily computed since there are only single word answers or executable code. I’m wondering how would you compute the EIG scores when the answer set is arbitrarily large. For example, for free form response generation where all answers are unique, and each answer can be arbitrarily long?

2. There is a typo in Line 900

---

> ### Author Response · Authors · 2024-11-19
> **Response to your review -- part 1**
>
> We thank the reviewer for their insightful comments and constructive feedback. Below, we address the questions and concerns raised by the reviewer.
>
> ---
>
> ### Weaknesses
>
> **W1) More challenging benchmarks**
>
> Thank you for raising this point. We would like to highlight that the primary focus of this paper is **task disambiguation**, specifically addressing uncertainty about the value of the ground-truth solution $h^∗$ that arises from the **inherent ambiguity of the initial problem statement** $\mathcal{S}^0$, rather than from the LLMs’s skill in solving a given class of problem. This distinction is clarified in the blue box before Section 2.1 of the manuscript. Ambiguity of a problem is defined with respect to the objective indicator function $1\\{h \vdash \mathcal{R}\\}$. A high uncertainty of the solution-generating distribution, $p_{\phi_h}$ does not indicate that the initial problem statement is itself ambiguous. Uncertainty of $p_{\phi_h}$ may arise both due to problem ambiguity and the problem solving agent's lack of skill in generating the correct answer to a well-specified problem.
>
> The benchmarks used in this paper have been carefully selected to balance the complexity of the problem-solving task with the degree of ambiguity in the problem statements, ensuring they are well-aligned with the research objectives:
>
> - **20 questions game:** While the problem-solving difficulty is intentionally low (the task is to identify an animal based on progressively refined characteristics), the ambiguity $\mathcal{S}^0$ is extremely high—any animal in the entire animal kingdom could initially be a valid solution. This setting emphasizes the core challenge of our work: resolving ambiguity through effective question generation.
> - **HumanEval Benchmark:** The problem-solving complexity is moderate, as the benchmark predominantly consists of simple coding problems**.**  At the same time, the ambiguity of these problems is significantly lower—for most problems, there exist only one of the problem statement $\mathcal{S}^0$.  However, as it has been demonstrated by Liu et al., some level of impreciseness of the problem statements is still present in this benchmark, leading to a few possible interpretations for a subset of these problems. We identify some of them in the Appendix, Table 4.  This makes HumanEval an appropriate benchmark for demonstrating the capabilities of our method in a domain where ambiguity and problem-solving complexity coexist.
>
> Regarding SWE-bench, while it is true that this benchmark is highly challenging, its design focuses on **problem-solving difficulty** rather than the **ambiguity of problem statements**. The majority of issues in SWE-bench are well-documented and precise, meaning the uncertainty in solving these tasks stems primarily due to the agent’s inability to solve complex problems. Out-of-the box LLM’s fail on this benchmark for ~95% of issues. The design of a sufficiently powerful solution generator ($h_i \sim p_{\phi_h}(\cdot \vert \mathcal{S})$) for this benchmark, is outsied of the scope of our work.  That said, our method is general, and given a sufficiently powerful solution generator, $h_i \sim p_{\phi_h}(\cdot \vert \mathcal{S})$, it has the potential to improve question/query/test-case generation for a variety of problem-solving scenarios.
>
> **UPDATE:** To test this claim, we have conducted a new set of experiments on the more challenging coding benchmark: APPS, which contains competition-level coding problems. We have selected a subset of problems which do not contain any input-output examples in their initial problem statement to ensure a sufficient level of ambiguity is present. Further, we have filtered this subset to only those on which GPT-4o-mini does not produce a correct solution zero-shot.
>
> Detailed results of these experiments are presented in Appendix D.2.1. Notably, the selected APPS subset has an average zero-shot accuracy of only ~35% with GPT-4o-mini, significantly lower than the ~70% on HumanEval, confirming that these problems are considerably more challenging. Despite this increased complexity, our EIG-based strategies consistently outperformed their non-EIG counterparts, with a substantial improvement of over 10 percentage points in accuracy. This demonstrates that our method remains effective even in more complex settings and underscores its robustness across varying levels of task difficulty and ambiguity. Thank you for your feedback, which enabled us to strengthen the validity and generality of our results.
>
> Reference:
>
> Liu, J., Xia, C., Wang, Y., & Zhang, L. (2023). Is Your Code Generated by ChatGPT Really Correct? Rigorous Evaluation of Large Language Models for Code Generation. https://arxiv.org/pdf/2305.01210

---

> > ### Author Response · Authors · 2024-11-19
> > **Response to your review -- part 2**
> >
> > ### Questions
> >
> >  **Q1) How would you compute the EIG scores when the answer set is arbitrarily large?**
> >
> > Thank you for your question. We would like to make a few remarks regarding the assumption on the finiteness of the answer set.
> >
> > - **Semantic equivalence.** We work under the assumption of noise-free answers and non-ambiguous questions, i.e. oracle answers $a$ about a solution $h \in \mathcal{H}$ to question $q$ are such that $(a, q)$ is compatible with $h$. This means that all possible answers to a single question $q$ must be semantically equivalent, regardless of their length in terms of the number of words. Consider the following question in the free-form version of the 20-questions game: “What is the colour of the animal’s fur?” For $h=$ “Polar bear”, the answers could be: “White”, “It is white”, “The animal’s fur is white”, etc. All answers are semantically equivalent.
> > - **Boundedness of the answer set.**  Letting $\mathcal{A}_q$ denote the set of semantically distinct answers to a question $q$. By definition, we must have that $|\mathcal{A}_q| \leq |\mathcal{H}|$, as answers correspond to partitions of the solution space.  At the extreme case of $|\mathcal{A}_q| = |\mathcal{H}|$ each answer uniquely identifies a hypothesis, leading to maximal information gain (see Corollary 1). However, this scenario is rare in practice and often undesirable due to the cognitive load imposed on the user.
> > - **Balancing information gain and user effort.** While questions with larger answer sets tend to provide higher information gain, they may impose a higher cognitive burden on users. One may argue that answering a question with more possible answers (e.g. an open-ended question) is more mentally demanding than selecting a single answer from a small, finite set. Working under the assumptions of BED (Rainforth el al.), we wish to maximise the expected information gain of a question, while minimising the cost of obtaining the answer. Under this view, questions with a small set of semantically-equivalent answers, but generating a balanced partitioning of the solution space, should be favoured (see blue box after Corollary 1).
> >
> > **Extension to free-form questions.** To compute EIG scores for free-form questions, we can extend our framework by relying on the idea of semantic entropy introduced by Kuhn et al., 2023. This approach would require a simple modification to our Algorithm 1—grouping simulated answers by their semantic equivalence rather than exact string match.  Techniques like semantic clustering based on bidirectional entailment (as outlined by Kuhn et al., 2023) can be used to map free-form answers to a finite set of equivalence classes. This ensures that the information gain is calculated based on meaningful distinctions between competing solutions.
> >
> > References:
> >
> > Kuhn, Lorenz, Yarin Gal and Sebastian Farquhar. “Semantic Uncertainty: Linguistic Invariances for Uncertainty Estimation in Natural Language Generation.”  https://arxiv.org/abs/2302.09664
> >
> > Rainforth, Tom, Foster, Adam, Ivanova, Desi R., & Bickford Smith, Freddie. "Modern Bayesian experimental design." Statistical Science, vol. 39, no. 1, 2024, pp. 100–114. Institute of Mathematical Statistics.
> >
> >
> >
> > **Typo in Line 900** Thank you for pointing this out. This has been now corrected in the updated version of our manuscript.
> >
> > ---
> >
> > We greatly appreciate the time and effort you’ve dedicated to reviewing our paper. We hope that the resulting additional experiments will strengthen the contributions of this paper and that our responses address your concerns satisfactorily.
> >
> > Thank you again for your valuable insights and for helping us improve the quality of this work.
> >
> > The authors of submission #11335

---

> > > ### Comment · Reviewer_xMi4 · 2024-11-25
> > >
> > > Thank you for your detailed responses and additional experiments, my concerns are mostly address and I decide to increase my score to 8.

---

> > > > ### Author Response · Authors · 2024-11-25
> > > > **Thank you**
> > > >
> > > > Dear reviewer,
> > > >
> > > > We sincerely thank the reviewer for taking the time to carefully consider our responses. We are grateful for the updated score and appreciate your acknowledgment of the contributions and improvements we made.
> > > >
> > > > Thank you for your continued engagement in the rebuttal process.
> > > >
> > > > Kind regards,
> > > >
> > > > The authors of submission #11335

---

### Author Response · Authors · 2024-11-19
**Global Response**

We would like to express our gratitude to all the reviewers for their constructive feedback and insights on our submission.

We are encouraged by the reviewers' recognition of our work's novelty and potential impact. We are pleased that the reviewers agree on the importance of the tackled problem of task ambiguity: “The proposed method of generating/selecting best clarification questions can have wide application to different tasks” (**xMiR**),  “Ambiguity and task ambiguity are increasingly important problems as more people use LLMs and AI systems (**qfUE**) .. The clarification-based approach they suggest has the potential to improve AI safety by reducing misunderstandings, and takes a human-centric approach to dealing with ambiguity.”,

We are also pleased that the reviewers recognised the clarity and rigour in presentation: “The work rigorously defines its problem and then offers a potential solution.” (**qfUE**), “The formulation of EIG scores for selecting the best clarification question is well designed.” (**xMi4**), “The paper provides a clear and formal definition of task ambiguity, which is crucial for framing and addressing the problem within a mathematical and computational context” (**HcvV).**

Regarding our empirical analysis, reviewers appreciated the inclusion of: “… real-world results on a naturalistic dataset in a coding domain, where many users actually use LLMs on a daily basis. This is a major strength of the paper. The authors also run across a large number of seeds and release code, making their approach more reproducible.” (**qfUE**),

We thank the reviewers for their positive feedback and appreciate the opportunity to clarify and enhance our manuscript based on the comments received. Below we outline the key actions taken in response to common concerns among the reviewers. Remaining queries of individual reviewers have been addressed in personalised responses.


### Summary of key actions taken

**Additional experimental results on more challenging coding problems.** Following the suggestion of reviewer **xMiR**, we have conduced another set of experiments on code-generation, but with a more challenging benchmark than HumanEval. We used a subset of problems of the APPS benchmark consisting of competition-level coding problems. On this subset, the average, zero-shot performance of GPT-4o-mini is less than 40% in comparison to to the near 70% accuracy on HumanEval, confirming the more challenging nature of this benchmark. Detailed results for all question generating strategies across four language models: GPT-4o-mini, GPT-3.5-turbo, Llama-3-70B, and Llama-3-8B, are available in Appendix D.2.1 in the updated version of the manuscript.  We also include a summary plot in the main body of the paper. Despite this increased complexity of the problems in the APPS benchmark, our EIG-based strategies consistently outperformed their non-EIG counterparts, with a substantial improvement of over 10 percentage points in accuracy. This demonstrates that our method remains effective even in more complex settings.

**Analysis of question acquisition costs.** Following the suggestion of reviewer **qfUE**, we have included a new section in the Appendix of the manuscript (Appendix E), where we provide a  comparison of the LLM sampling costs associated with the competing question generating strategies. As noted in the final section of our paper, the question-generating strategies presented in this work require an increased number of LLM calls compared to the baselines. However, in line with the assumptions commonly made in Bayesian Experimental Design, we take the stance that the computational load required to select the optimal query is negligible compared to the value of acquiring information that reduces problem ambiguity. We anticipate this assumption will become more valid over time as technology advancements lower the costs of LLM token generation, thereby enhancing the importance of efficient information acquisition strategies.

**Illustrative example.** Following the suggestion of reviewer **HccV**, we included a concrete example of an ambiguous problem statement in the main body of the text. This is presented as a continuation of Example 1. We hope that this addition helps the readers grasp the key concepts discussed in the theoretical section of the paper.

### Updated manuscript

Based on the helpful and detailed feedback of the reviewers we have updated our manuscript and uploaded the revised version. All changes made are highlighted with a blue text color. We hope that the proposed updates improve the paper’s clarity, ease of understanding, and that they address the reviewers’ questions and concerns.

---

We are grateful for the reviewers' feedback, which helped us improve the presentation of our work. We are open to further discussions to clarify any aspects of our submission.

Kind regards,

The authors of submission #11335

---

### Meta-Review · Area_Chair_cjsz · 2024-12-23

**Metareview:**

The paper presents a novel Bayesian Experimental Design (BED)-based approach for resolving task ambiguity in LLMs by generating clarifying questions using Expected Information Gain (EIG). The approach is both theoretically and practically significant, offering improvements in clarity, task efficiency, and robustness. The reviewers praised the paper for its clear presentation, strong empirical validation on challenging benchmarks like APPS, and broad applicability to real-world tasks.

Initially, reviewers raised concerns about the simplicity of benchmarks, computational costs, and clarity of examples. These were thoroughly addressed by the authors through additional experiments, a detailed cost analysis, and improved explanations in the revised manuscript. As a result, reviewer scores were revised upwards, with two reviewers assigning a rating of 8 and one a 6, reflecting overall strong support.

The paper's contributions and the authors' responsiveness to feedback make it a compelling candidate for acceptance.

**Additional Comments On Reviewer Discussion:**

The rebuttal addressed concerns with new experiments, examples, and cost clarifications, improving clarity and supporting the recommendation for marginal acceptance.

---

### Decision · Program_Chairs · 2025-01-22

Accept (Spotlight)